# Chromatogram libraries improve peptide detection and quantification by data independent acquisition mass spectrometry

Brian C. Searle[1,2], Lindsay K. Pino[1], Jarrett D. Egertson[1], Ying S. Ting [1], Robert T. Lawrence[1], Brendan X. MacLean[1], Judit Villén[1] & Michael J. MacCoss [1]

Data independent acquisition (DIA) mass spectrometry is a powerful technique that is improving the reproducibility and throughput of proteomics studies. Here, we introduce an experimental workflow that uses this technique to construct chromatogram libraries that capture fragment ion chromatographic peak shape and retention time for every detectable peptide in a proteomics experiment. These coordinates calibrate protein databases or spectrum libraries to a specific mass spectrometer and chromatography setup, facilitating DIA-only pipelines and the reuse of global resource libraries. We also present EncyclopeDIA, a software tool for generating and searching chromatogram libraries, and demonstrate the performance of our workflow by quantifying proteins in human and yeast cells. We find that by exploiting calibrated retention time and fragmentation specificity in chromatogram libraries, EncyclopeDIA can detect 20–25% more peptides from DIA experiments than with data dependent acquisition-based spectrum libraries alone.

[1] Department of Genome Sciences, University of Washington, Seattle, WA, USA. [2] Proteome Software, Portland, OR, USA. Correspondence and requests for materials should be addressed to M.J.M. (email: maccoss@uw.edu)

Over the past two decades the continued refinement of proteomics methods using liquid chromatography (LC) coupled to tandem mass spectrometry (MS/MS) has enabled a deeper understanding of human biology and disease[1,2]. Recently data independent acquisition[3,4] (DIA), in which the mass spectrometer systematically acquires MS/MS spectra irrespective of whether or not a precursor signal is detected, has emerged as a powerful alternative approach to data dependent acquisition[5] (DDA) for proteomics experiments. In current DIA workflows, instrument cycle is structured such that the same MS/MS spectrum window is collected every 1–5 s, enabling quantitative measurements using fragment ions instead of precursor ions. This approach produces data analogous to targeted parallel reaction monitoring (PRM), except instead of targeting specific peptides, quantitative data is acquired across a predefined mass to charge ($m/z$) range. One trade-off is that to cover the $m/z$ space where the majority of peptides exist, the mass spectrometer must be tuned to produce MS/MS spectra with wide precursor isolation windows that often contain multiple peptides at the same time. These additional peptides produce interfering fragment ions, and database search engines for DDA that rely on a precursor isolation window of at most a few daltons can struggle to detect the signal for a particular peptide from that background interference. The PAcIFIC approach[6] attempts to overcome this difficulty by using multiple gas-phase fractionated injections of the same sample to increase precursor isolation at the cost of both sample and instrument time.

Spectrum-centric tools[7,8] attempt to deconvolve peptide signals from DIA data by time aligning elution peaks for both fragment and precursor ions. In contrast, peptide-centric tools analyze DIA measurements to look for individual peptides across all spectra in a precursor isolation window. Spectrum library search tools for DIA data[9–12] use fragmentation patterns and relative retention times from previously collected DDA data. Other tools such as PECAN[13] query DIA data using just peptide sequences and their predicted fragmentation pattern without requiring a spectrum library. While library searching can achieve better sensitivity than PECAN, the approach is limited to detecting only analytes represented in the library. In addition, the quality of library-based detections is only as strong as the quality of the library itself. Because mapping fragmentation patterns and retention times across instruments and platforms is difficult, many researchers prefer to simultaneously acquire both DDA and DIA data from their samples[14,15]. While this implicitly increases the acquisition time and sample consumption, it becomes possible to detect peptides using the DDA data while making peptide quantitation measurements using the DIA data. However, detection sensitivity is inherently limited to that of the DDA data.

Typically tens to hundreds of biological samples are processed and analyzed using LC-MS/MS in quantitative proteomics experiments. The regularity of DIA allows researchers to make peptide detections in one sample and transfer those detections to other samples[16]. Here, we extrapolate this concept by collecting certain runs where data acquisition is tuned to improve peptide detection rates, while collecting other runs with a focus on quantification accuracy and throughput. These runs can be searched using either a typical DDA spectrum library-based workflow or a pure DIA workflow using PECAN, or spectrum-centric search methods based on DIA-Umpire[8] or Spectronaut Pulsar. Results from runs dedicated to peptide detection are formed into a DIA-based chromatogram library. In a chromatogram library, we catalog retention time, precursor mass, peptide fragmentation patterns, and known interferences that identify each peptide on our instrumentation within a specific sample matrix. Furthermore, we report the development EncyclopeDIA, a library search engine that takes advantage of chromatogram

libraries, and we demonstrate a substantial gain in sensitivity over typical DIA and DDA workflows. This tool is instrument vendor neutral and available as an open source project with both a GUI and command line interface.

## Results

**The EncyclopeDIA workflow.** EncyclopeDIA is comprised of several algorithms for DIA data analysis (Fig. 1b) that can search for peptides using either DDA-based spectrum libraries or DIA-based chromatogram libraries. In addition, the EncyclopeDIA executable contains the Walnut search engine, which is a performance optimized re-implementation of the PECAN algorithm[13] to search protein sequence FASTA databases (see Supplementary Note 1 for further details). The algorithms in the EncyclopeDIA workflow are described in full detail in the Methods section. Briefly, the EncyclopeDIA workflow starts with reading raw MS/MS data in mzML files into an SQLite database designed for querying fragment spectra across precursor isolation windows. If fragment spectra are collected using overlapping windows, they are deconvoluted on the fly during file reading. Libraries are read as DLIB (DDA-based spectrum libraries) or ELIB (DIA-based chromatogram libraries). EncyclopeDIA determines the highest scoring retention time point corresponding to each library spectrum (as well as a paired reverse sequence decoy) using a scoring system modeled after the X!Tandem HyperScore[17]. Fifteen auxiliary match features (not based on retention time) are calculated at this time point. These features are aggregated and submitted to Percolator 3.1[18], a semi-supervised SVM algorithm for interpreting target/decoy peptide detections, for a first pass validation. EncyclopeDIA generates a retention time model from peptides detected at 1% FDR using a non-parametric kernel density estimation algorithm that follows the density mode across time. Any target or decoy peptide in the feature set that does not match the retention time model is reconsidered up to five times until we find a highest scoring retention time point that matches the model. The retention time-curated feature sets are submitted to Percolator for final pass validation at 1% peptide FDR.

**Chromatogram library generation.** Gas-phase fractionated DIA uses multiple injections with data acquisition methods that are tiled to span different precursor isolation windows[6]. With modern instrumentation it is possible to collect near proteome-wide DIA measurements with equivalently narrow precursor isolation to DDA using as few as six gas-phase fractionated injections. Previously we have shown that this type of DIA experiment can produce substantially richer peptide detection lists than similarly acquired DDA experiments[13]. In addition, it is much easier to detect low abundance peptides from gas-phase fractionated DIA using library search engines or database search engines than when searching wide-window DIA runs, which attempt to collect near proteome-wide measurements with a single injection. However, this strategy is impractical in both total instrumentation time and sample requirements to be performed for large quantitative experiments. We propose an approach to collecting DIA data using chromatogram libraries that leverages the deep sampling of gas-phase fractionated DIA while still maintaining high throughput (Fig. 1a).

In addition to collecting wide-window DIA experiments of each biological sample, we also collect narrow-window gas-phase fractionated DIA runs of pooled subaliquots of those samples. We detect peptides from the resulting narrow precursor isolation windows using library search engines (such as EncyclopeDIA) or DIA-specific database search engines (such as Walnut). To generate a chromatogram library, we catalog the retention time,

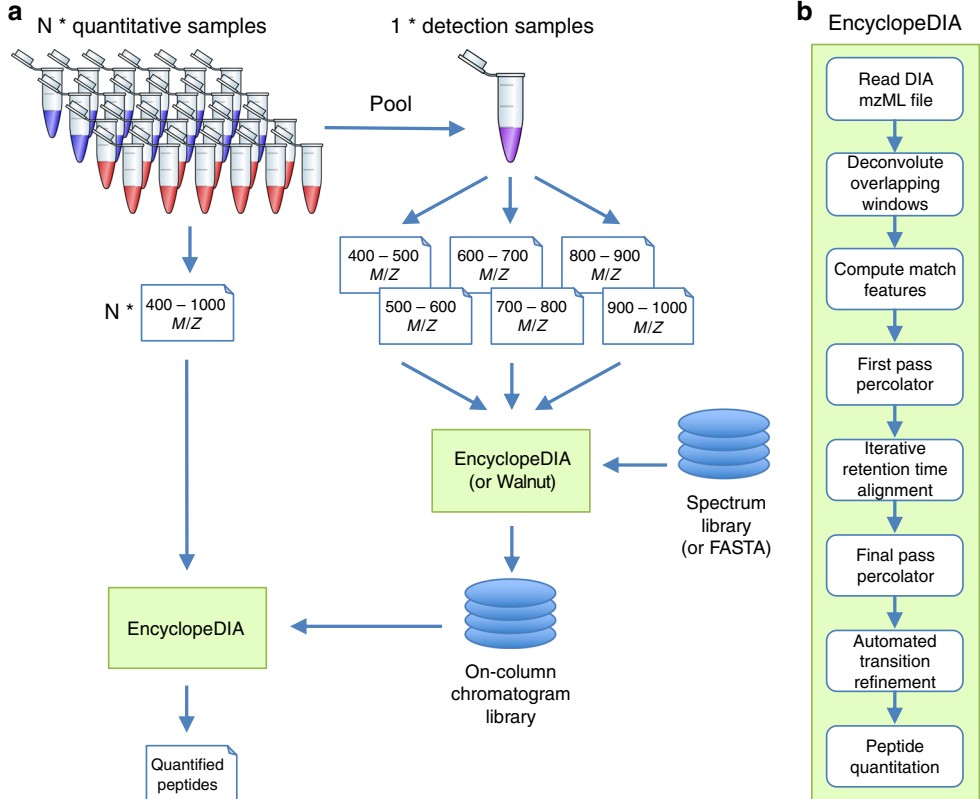

**Fig. 1** An approach for quantifying peptides with chromatogram libraries. **a** The chromatogram library generation workflow. Briefly, in addition to collecting wide-window DIA experiments on each quantitative replicate, a pool containing peptides from every condition is measured using several staggered narrow-window DIA experiments. After deconvolution, these narrow-window experiments have 2 m/z precursor isolation, which is analogous to targeted parallel reaction monitoring (PRM) experiments, except effectively targeting every peptide between 400 and 1000 m/z. We detect peptide anchors from these experiments using either EncyclopeDIA (searching a DDA spectrum library) or PECAN/Walnut (using a protein database) and chromatographic data about each peptide is stored in a chromatogram library with retention times, peak shape, fragment ion intensities, and known interferences tuned specifically for the LC/MS/MS setup. EncyclopeDIA then uses these precise coordinates for m/z, time, and intensity to detect peptides in the quantitative samples. **b** The EncyclopeDIA algorithmic workflow for searching spectrum and chromatogram libraries. After reading and deconvoluting DIA raw files, EncyclopeDIA calculates several retention time independent feature scores for each peptide that are amalgamated and FDR corrected with Percolator. Using high confidence peptide detections, EncyclopeDIA retention time aligns detections to the library, determines the retention time accuracy, and reconsiders outliers. After a second FDR correction with Percolator, EncyclopeDIA autonomously picks fragment ion transitions that fit each non-parametrically calculated peak shape and quantifies peptides using these ions

peak shape, fragmentation patterns, and known interferences of detected peptides filtered to a 1% global peptide FDR. With DIA experiments we expect interference, so rather than removing impure library spectra and spectra that contain multiple peptides[19], we simply only retain +1H and +2H fragment ions for expected B-type and Y-type ions. Due to gas-phase fractionated tiling, each peptide is only represented in the narrow-window data once, which eliminates the need for spectrum averaging[19] or best spectrum selection[20] typically used by DDA-based library curation tools. In the chromatogram library for each peptide we retain only the highest scoring charge state (as determined by Percolator) to limit the search space.

Chromatogram libraries implicitly contain a subset of the peptides found in DDA-based spectrum libraries, but the peptides they do contain have chromatographic and fragmentation data calibrated specifically to DIA experiments on that instrumentation platform. One limitation is that peptides that cannot be detected in narrow-window DIA runs of the pooled sample will not be searched for in subsequent runs. We feel that very few quantitatively reliable peptides will be detectable in the wide-window data that are not also detectable in the narrow data and that the smaller search space represented by chromatogram libraries can increase the significance of peptide detections[21]. In

cases where rare variants are important to a study or if samples are likely to represent very disparate proteomes, it may also be possible generate chromatogram libraries from multiple batches of narrow-window acquisitions from different sample pools.

In this study we generated a chromatogram library using peptides derived from HeLa S3 cell lysates. First we used Skyline to assemble a HeLa-specific DDA-based spectrum library containing 166.4k unique peptides (representing 9947 protein groups) from 39 raw files acquired for other experiments. These files were collected from SCX and high-pH reverse phase fractions acquired with a Thermo Q Exactive tandem mass spectrometer using multiple HPLC gradients to vary the local peptide matrix. Using this library as a starting point, we constructed a chromatogram library from six gas-phase fractionated DIA runs with 52 overlapping 4 m/z-wide windows. We collected these runs with a Thermo Q-Exactive HF tandem mass spectrometer using a 90 min linear gradient. After overlap deconvolution, these experiments produced 300 2 m/z-wide windows, which is analogous to if we had conducted targeted PRM acquisition except that we are targeting all precursors between 400.43 and 1000.70 m/z. Following the scheme in Fig. 1a, we searched the narrow-window data against a HeLa-specific DDA spectrum library (166k unique peptides), producing a

chromatogram library containing 99.6k unique peptides, and an analogous search against the Pan-Human spectrum library[22] (159k unique peptides) producing a chromatogram library containing 91.1k unique peptides. We also produced a third library containing 53.2k unique peptides using Walnut to detect peptides directly from the narrow-window DIA data using a Uniprot Human FASTA database. The difference in chromatogram library size by searching DDA-based spectrum libraries with EncyclopeDIA or a FASTA database (1143k unique +2H/+3H tryptic peptides) with Walnut is in part because the spectrum library represents a more targeted search space, while additionally including expected post-translationally modified (oxidized and acetylated) peptides, as well as peptides with multiple missed cleavages and expected +4H/+5H/+6H peptides.

**Comparison of spectrum and chromatogram library searches.** We evaluated the chromatogram library strategy using peptides derived from a HeLa S3 cell lysate as a representative high-complexity proteome. In addition to generating the library, we also collected triplicate wide-window DIA runs with 52 overlapping 24 m/z-wide windows from the same sample using the

same 90 min linear gradient. We found an average of 72.3k peptides when searching against the chromatogram library constructed using a HeLa-specific DDA-based spectrum library. Corroborating experiments from other groups[23,24], with DIA we can detect up to 2x more peptides than our benchmark top-20 DDA experiments (Supplementary Figure 1). While Bruderer et al.[23] found a significant performance drop when comparing results from previously acquired global libraries (such as the Pan-Human library[22]) to experiment-specific DDA spectrum libraries, we did not find a similar drop when searching chromatogram libraries generated from the Pan-Human library. This result indicates that our approach enables the reuse of previously acquired global libraries intended as community standards without requiring generating experiment-specific DDA libraries. In a sense, chromatogram libraries provide a calibration step that substitutes the data in DDA spectrum libraries or fragmentation models in database search engines for DIA-specific fragmentation and HPLC/column-specific retention times. Despite this increased detection rate, we still find that DIA produces more consistent results compared to DDA, as indicated by the overlap in peptide detections between triplicate injections (Fig. 2b, c). This agrees with previous reports that DIA quantification is both more uniform[13] and more accurate[8,10,12]. While other library

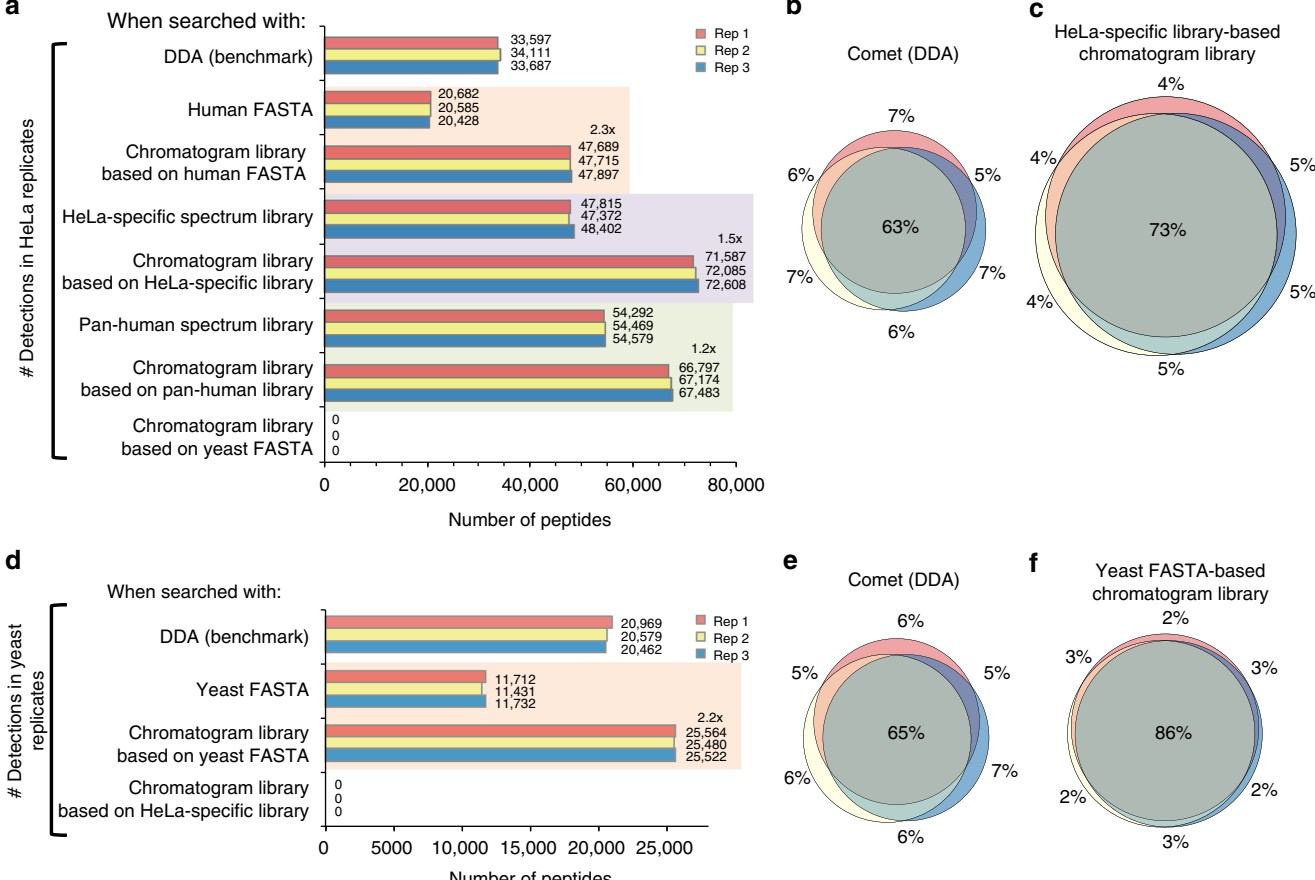

**Fig. 2** Untargeted peptide detections using DDA and DIA. We used EncyclopeDIA to search chromatogram and spectrum libraries, while we used Comet and Walnut to search DDA and DIA data directly using FASTA protein databases. Every search was performed independently without any run-to-run alignment. **a** The number of peptide detections at 1% peptide FDR in triplicate HeLa injections. Orange shaded areas indicate pairwise comparisons of FASTA searches versus FASTA-based chromatogram library searches. Purple (or green) shaded areas indicate pairwise comparisons of searches of a cell line-specific DDA library (or the Pan-Human DDA library) versus a chromatogram library derived from that DDA library. **b** The overlap in HeLa S3 peptide detections between replicates using DDA searched by Comet and **c** using DIA searched by EncyclopeDIA where the size of Venn diagram circles in HeLa analyses are consistent with the number of detections. **d** The number of peptide detections at 1% peptide FDR in triplicate BY4741 yeast injections. **e** The overlap in yeast peptide detections between replicates using DDA searched by Comet and **f** using DIA searched by EncyclopeDIA where the size of circles are consistent with the number of yeast peptide detections

search tools such as Skyline[25] cannot make use of all chromatogram library features, we find that Skyline still produces higher detection rates when searching chromatogram libraries as compared to both the HeLa-specific and Pan-Human DDA-based spectrum libraries (Supplementary Figure 2).

We also evaluated the creation of chromatogram libraries using a DIA-only workflow. Using this approach, we were able to detect an average of 20.6k peptides from the Uniprot Human FASTA database using Walnut, or approximately 0.6× of the detections found by top-20 DDA. In contrast, we found an average of 47.8k peptides (2.3× increase) when we searched the Walnut-based chromatogram library with EncyclopeDIA (Fig. 2a), or approximately 1.4× more than DDA. These results agree with previous work[13] showing that Pecan does not perform as well as DDA when searching wide-window runs, but typically outperforms DDA when searching gas-phase fractionated runs. Interestingly, the DIA-only workflow found nearly an equal number of peptides compared to searching the 39 injection HeLa-specific DDA-based spectrum library, while requiring only an additional six library-building injections. Confirming these results, we performed the same analysis using a yeast cell lysate and found similar improvement rates when comparing Walnut versus Encyclope-DIA using a Walnut-based chromatogram library (2.2× increase, Fig. 2d), or 1.2× more than top-20 DDA. Here we observe more modest gains over DDA experiments, which likely reflects the lowered proteomic complexity of yeast versus human cells and is echoed in the tight overlap (86%) between triplicate DIA injections versus DDA (Fig. 2e, f). As is possible with any computational strategy that incorporates machine learning, we were concerned with the potential for overfitting that might manifest in over exaggerated peptide detection rates. To answer this question we searched the HeLa wide-window DIA data using the yeast chromatogram library (and vice versa) to verify that we see a negative result when searching the wrong library. As expected this result (Fig. 2a, d) produced zero peptide detections that passed a 1% peptide FDR threshold.

We also find that DIA analysis with chromatogram libraries is more sensitive at detecting low abundance proteins at a 1% protein FDR. Using tandem affinity purification tagging and quantitative Western blots, Ghaemmaghami et al.[26] quantified 3868 yeast proteins with more than 50 estimated copies per cell. In this study we replicated strain and growing conditions as closely as possible to use their measurements as an independent benchmark. While both DDA and DIA confidently detect the majority of proteins at levels above $10^4$ copies per cell, DIA outperforms DDA by 49% with proteins estimated to have between $10^3$ and $10^4$ copies per cell and by 2× with proteins estimated between $10^2$ and $10^3$ copies per cell (Fig. 3).

**Retention time and fragmentation pattern calibration**. One of the primary reasons on-column chromatogram libraries improve performance is that they exploit within batch retention time reproducibility. Accurate retention time filtering is an important consideration when analyzing high-complexity proteomes with DIA, and virtually all DIA library search engines make use of this data. Retention times in aggregate spectrum libraries are typically

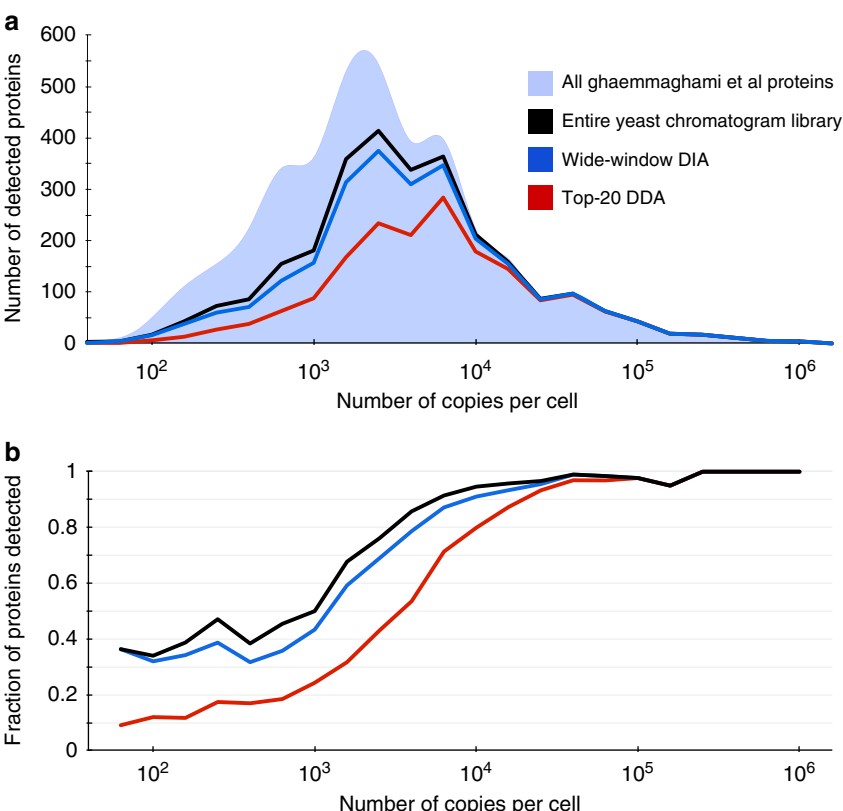

**Fig. 3** Protein detection rates scale with abundance. The **a** number and **b** fraction of proteins detected in yeast at different orders of magnitude of abundance. Ghaemmaghami *et al* comprehensively estimated protein copies per cell in yeast (light blue area, 3868 total proteins) using high-affinity epitope tagging. While top-20 DDA (red line, 1798 total proteins) can measure some low abundant proteins at 1% protein-level FDR, the strategy only detected 48% of mid-range proteins with estimated copies per cell between $10^3$ and $10^4$. In contrast, at 1% protein-level FDR, wide-window DIA using a Walnut-based chromatogram library (blue line, 2519 total proteins) detected 71% of these proteins and overall recapitulated 91% of proteins found in the entire Walnut-based chromatogram library (black line, 2754 total proteins)

derived by linearly interpolating multiple DDA data sets to a known calibration space (such as that defined by the iRT standard[27]), which enables retention times to be comparable from run to run, or even across platforms. However, these measurements usually contain some wobble due to errors introduced by assuming a linear fit. Bruderer et al.[28] improve upon this strategy with high-precision iRT fitting using a non-parametric curve fitting approach for hundreds or thousands of peptides, and EncyclopeDIA uses an analogous kernel density estimation approach to fit retention times between wide-window DIA results and retention times in libraries. Figure 4a shows a typical spread of retention times in EncyclopeDIA detected peptides using a DDA spectrum library, which is 95% accurate within a spread of 5.1 min (Fig. 4c). In comparison, Fig. 4b shows the typical spread of retention times in the chromatogram library, which is 95% accurate within 21 s (Fig. 4d). This tightening of retention time

accuracy is due to the fact that chromatogram libraries are collected on the same column as the wide-window acquisitions. Even if efforts are made to keep packing material, length, and gradient consistent, the dramatic gains in retention time accuracy with chromatogram libraries reflect variations that are difficult to control for, including packing speeds, pressures, and pulled tip orifice shapes. In addition, we find that DDA fragmentation patterns (Fig. 4e) are often somewhat different than those collected in DIA experiments (Fig. 4f). While DDA instrument methods usually tune MS/MS collision energies to the precursor charge and mass, some of this variation is likely due to fixed assumptions in charge states and precursor masses required by DIA methods when multiple precursors must be fragmented at the same time. These two factors appear to have relatively equal and orthogonal improvements over searching DDA spectrum libraries (Supplementary Figure 3).

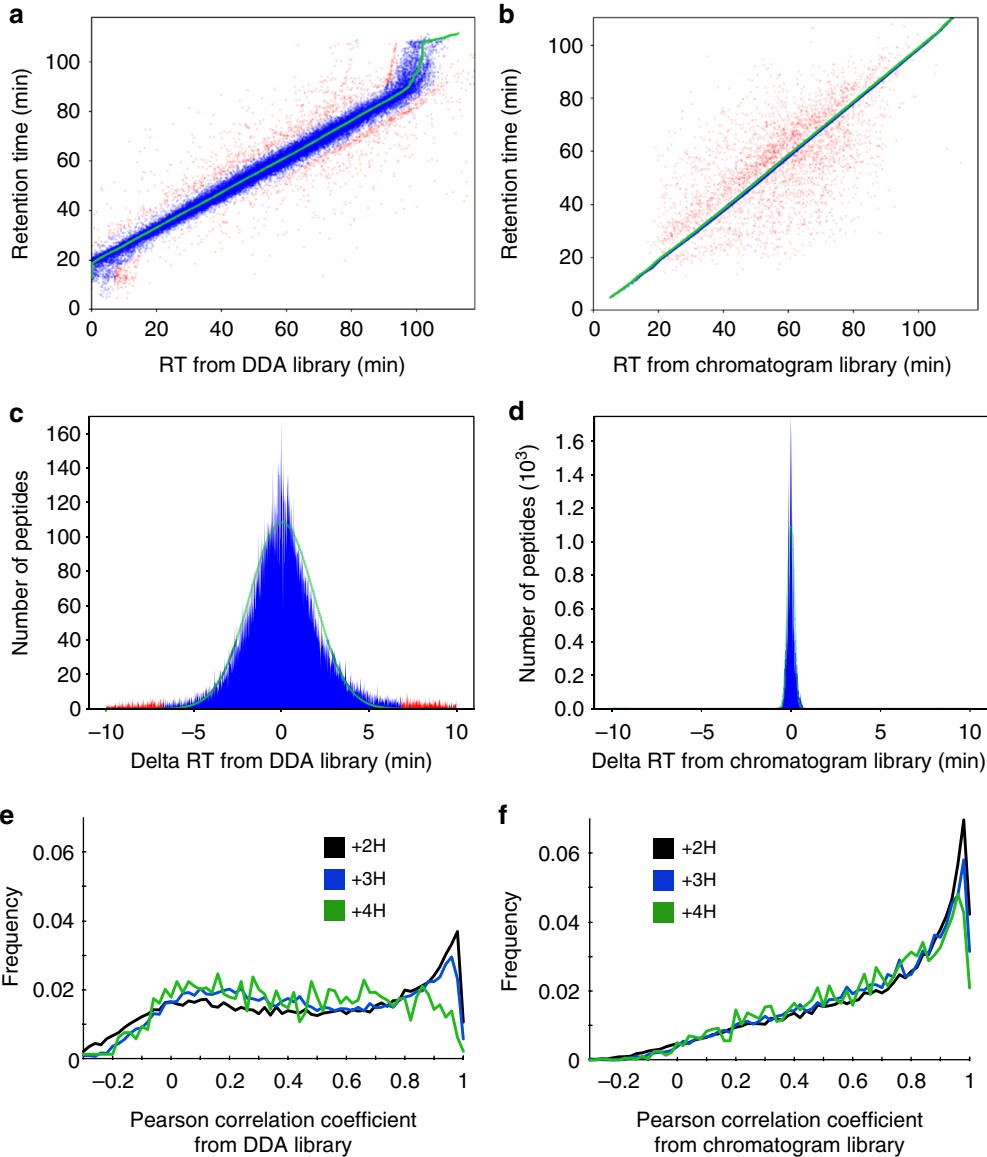

**Fig. 4** Retention time and fragmentation accuracy of spectrum and chromatogram libraries. Scatterplots comparing retention times (RT) from the **a** DDA spectrum library and the **b** DIA chromatogram library to those from in a single HeLa DIA experiment. Each point represents a peptide, where blue peptides fit the retention time trend (green) within a Bayesian mixture model probability of 5% and red peptides are outliers (see Methods section for more details). **c** Delta RTs in the DDA spectrum library are 95% accurate to a window of 5.1 min, while **d** retention times in the chromatogram library are 95% accurate to 21 s. **e** The distribution of Pearson correlation coefficients between spectra in the DDA spectrum library and those detected from a single HeLa DIA experiment shows charge state bias, while **f** the distribution of correlation coefficients between spectra in the DIA chromatogram library and those from the same experiment shows much less bias

A subtle issue with DIA library searching when using generalized spectrum libraries is that many peptides generate the same fragment ions, either because of sequence variation, paralogs, or modified forms. While EncyclopeDIA attempts to control for this using background ion distributions to predict interference likelihoods, sequence variation due to homology or single nucleotide polymorphisms can be unintentionally detected as the wrong peptide sequence in certain circumstances. For example, a sequence variation of a valine to an isoleucine is relatively common, and the mass shift of a methyl group ($+14/Z$) will often place both peptides inside the same precursor isolation window when $Z$ is 2 or greater. Using chromatogram libraries can provide some protection against these issues because the initial searches to generate the libraries are performed using narrow (2 m/z) precursor mass windows, and subsequent wide-window searches benefit from precise retention time filtering. Additionally, EncyclopeDIA requires at least 25% of the primary score to come from ions that indicate the modified form to detect modified peptides when modified/unmodified peptide pairs fall in the same precursor isolation window (e.g., methionine oxidation).

**Peptide and protein quantitation.** Automated interference removal is an important aspect to analyzing wide-window DIA data. SWATHProphet[29] attempts to solve this by comparing relative fragment intensities in spectrum libraries to those found in the DIA data, while mapDIA[30] computes the correlation between every pair of fragment ions to identify outliers. We present an algorithm for automated transition refinement to remove fragment ion interference and alleviate the need for manual curation (see Methods section for further details). In short, after unit area normalizing all transitions assigned to a single peptide (Supplementary Figure 4a), we determine the shape of the peak as the median normalized intensity at each retention time point (Supplementary Figure 4b). Transitions that match this peak shape with Pearson's correlation scores > 0.9 are considered quantitative (Supplementary Figure 4c). We find that over 81% of peptides can be quantified with at least three transitions (Supplementary Figure 5a) and that the transitions picked by our approach produce reproducible quantitative measurements between technical replicates in HeLa experiments (Supplementary Figure 5b and c).

Combining peptide detections across multiple samples often increases false discoveries because false detections are usually found only in individual runs. To combat this, we recalculate global peptide FDR across all experiments in a study[31] with Percolator and generate parsimonious protein detection lists that are also filtered to a 1% global protein FDR. We use cross-sample retention time alignment[16] to help quantify peptides that are missing in specific samples. After filtering peptides based on coefficient of variance and measurement consistency, we estimate protein quantities by summing fragment ion intensities across only sequence-unique peptides assigned to those proteins. Using a similar strategy to LFQbench[32], we validated the quantitative accuracy of protein-level measurements with triplicate experiments of five different mixtures of yeast and HeLa proteomes at expected concentrations (Fig. 5). In these mixtures we detected 2563 yeast proteins that passed a 1% global protein FDR threshold. Of these, we found that 2018 yeast proteins produced at least three quantitative transition ions without interference, had <20% study-wide CVs, and were measured in every replicate in pure yeast experiments. While at first these detection and quantification criteria may seem stringent compared to typical proteomics experiments, narrowing our focus to confident measurements produced quantitative ratios that closely adhered to the expected mixture ratios, especially with regards to small fold changes. We employed these methods and filtering criteria to study the effect of serum starvation in human cells.

**Global proteomic changes from serum starvation.** Serum starvation is a common step in signal transduction studies as serum contains several cytokines and growth factors that can confound signaling levels. It is commonly thought that serum starvation suppresses basal activity by reducing signaling activity that effectively resets cells to G0/G1 resting phase[33], although more recent experiments[34,35] suggest otherwise. Serum starvation protocols vary widely from 2 to 24 h, and this time frame is long enough to produce changes in protein levels resulting from transcriptional regulation. These changes are a source of variation that can have serious consequences when comparing between studies.

We designed a DIA quantitative experiment to map how the proteome of HeLa cells changes in response to serum starvation

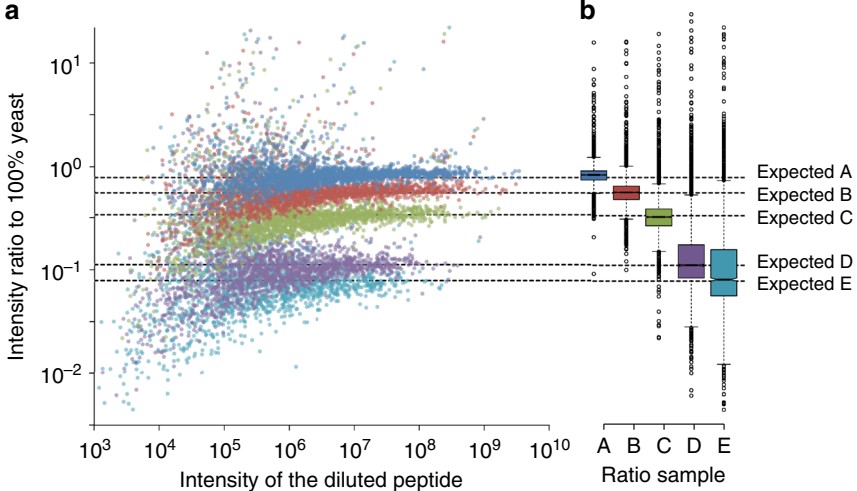

**Fig. 5** Quantitative accuracy in mixed proteomes. **a** Quantitative ratios of 2018 yeast proteins spiked into a HeLa background at five different concentrations and measured in triplicate. Each point indicates the average protein ratio relative to 100% yeast. **b** Boxplots showing the spread of the ratio measurements where boxes indicate medians and interquartile ranges, and whiskers indicate 5 and 95% values. The expected dilution ratios for samples A, B, C, D, and E are 78, 56, 34, 11, 7.8% yeast

over time. We selected six starvation times to match commonly used protocols and generated six biological replicates per condition. We collected all the DIA runs with the same mass spectrometer and chromatographic conditions. Of the 99.6k unique peptides in our chromatogram library, we recapitulated 93.5k unique peptides from 6,802 protein groups at a 1% global protein FDR threshold. As above, 48.6k peptides (from 5,781 protein groups) produced at least three quantitative transition ions without interference, had <20% study-wide CVs, and were measured in every replicate of at least one time point.

Using EDGE[36] we found 1097 protein groups in the HeLa proteome that changed significantly over time at a *q*-value < 0.01 (Supplementary Data 1). The temporal starvation profiles of these proteins fell into five groups (Fig. 6) where the majority changing proteins increased in abundance. Several of these proteins are involved in expected pathways such as cell cycle regulation (GO enrichment FDR = 0.011), metabolism (GO enrichment FDR = 0.011), and ubiquitination regulation (GO enrichment FDR = 0.018). One advantage of our method is that quantitation is performed by summing peaks from several low interference fragment ions, which allows us to accurately quantify small changes. For example, we found that all eight of the observed components of the nuclear proteasome increased significantly by ~25% (Supplementary Figure 6), which indicates nuclear maintenance consistent with G0/G1 resting phase.

We also observed significant regulation of the abundance of 39 kinases and 7 phosphatases (Supplementary Figure 7). In particular, we found that EGFR levels increased by 30% over a 24 h serum starvation time course (Supplementary Figure 8),

effectively sensitizing HeLa to the growth factor EGF. To confirm these experiments, we monitored relative changes in the phosphoproteome of four HeLa biological replicates after EGF stimulation at two common serum starvation times: 4 and 16 h. We found that while phosphopeptide measurements at both time points directionally agreed, some phosphopeptide responses to EGF were stronger when cells were starved for 16 h compared to when starving for only 4 h (Supplementary Figure 9). This increase corroborated our observation that EGFR protein levels increased from 4 to 16 h of starvation. These protein and phosphopeptide-level changes underline a potentially significant source of variation when comparing phosphorylation signaling studies.

## Discussion

We have demonstrated an experimental strategy that enables comprehensive detection of peptides and proteins using chromatogram libraries. These libraries can be seeded either with a DDA spectrum library or generated in a DIA-only mode using Walnut for initial peptide searches. Finally, we showed that at the cost of only six additional narrow-window DIA runs, both of these strategies are more sensitive and reproducible relative to comparable DDA experiments. While this approach may be unrealistic for one-off experiments, we feel that in most quantitative proteomics studies the addition of these runs are a minor cost in exchange for a significant increase in sensitivity.

One important limitation of our method is that each chromatogram library is tuned for a specific mass spectrometer and chromatographic set up. In particular, we have observed that with the hand-pulled and packed columns used here, there is significant retention time variation between replicates run on different columns, even if effort is made to ensure column consistency. We hypothesize that minor variations in packing speeds, packing pressures, tip shapes, and column lengths can affect elution times and even peptide retention time ordering. This issue may be mitigated by acquiring a new library after a column change and retention time aligning the libraries to ensure consistency. Future work remains to model these minor retention time shifts.

Another important consideration is library quality. All library searching strategies assume that entries in the library are correctly identified and consequently false positives in the library can be propagated as true positives by target/decoy analysis[37]. This concern is potentially compounded in our approach, which can include up to two levels of library creation. Further work is necessary to improve FDR estimates for library searching in DIA experiments. In the meantime, we feel orthogonal filtering strategies are necessary to maintain conservative peptide detection lists. In addition to retention time fitting and 1% protein-level FDR filtering, in this work we require a minimum of three interference-free transitions and impose stringent measurement reproducibility requirements for peptides to be considered quantitative.

We have observed a complementarity of DDA and DIA through the use of building spectrum libraries to seed chromatogram libraries. Here the stochasticity of DDA sampling when coupled with offline peptide separation methods such as SCX fractionation can be exploited as a benefit in that only one observation of a peptide is necessary for inclusion in the library. With human samples, libraries constructed using previously recorded retention times and fragmentation patterns contained nearly twice the peptides as those constructed without prior knowledge. However, PECAN/Walnut can build on that knowledge by detecting peptide sequence variants illuminated by whole exome sequencing[13], and we are exploring ways of

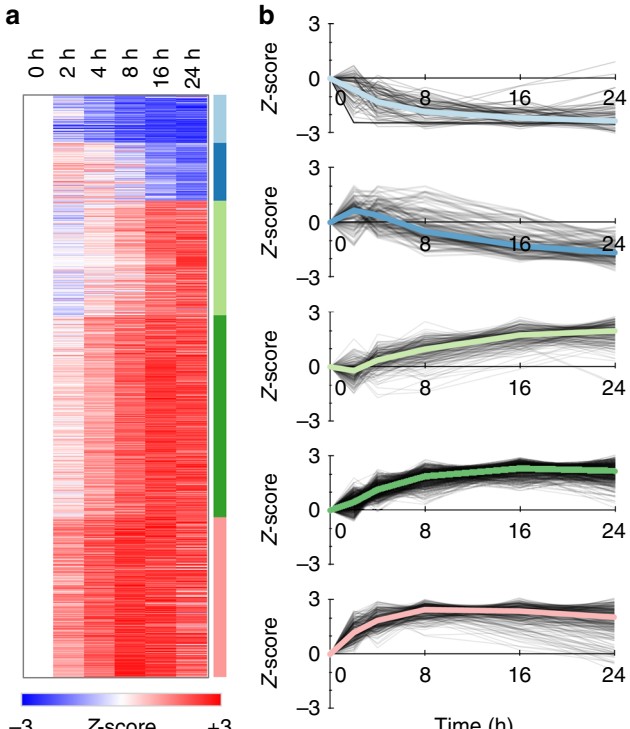

**Fig. 6** Protein quantification changes following serum starvation. **a** Heatmap of 1097 proteins found to be quantitatively changing at a *q*-value < 0.01 in HeLa. Colors are *Z*-score normalized and indicate the number of standard deviations away from the level at time 0. **b** Protein changes grouped into five K-means clusters (see Supplementary Figure 13 for more details) showing separation between fast responding proteins (light blue, dark green, and pink) and delayed responses (dark blue, light green)

generating chromatogram libraries that incorporate both pieces of data.

## Methods

**HeLa cell culture and sample preparation.** HeLa S3 cervical cancer cells (ATCC) were cultured at 37 °C and 5% $CO_2$ in Dulbecco's modified Eagle's medium (DMEM) supplemented with L-glutamine, 10% fetal bovine serum (FBS), and 0.5% strep/penicillin. Six cell culture replicates were grown to approximately a 50% density in 6-well plates prior to FBS starvation staggered for 24, 16, 8, 4, 2, and 0 h (one time point in each well, one plate per replicate). At the 0 h time point cells were quickly washed three times with refrigerated phosphate-buffered saline (PBS) and immediately flash frozen with liquid nitrogen. Frozen cells were lysed in a buffer of 9 M urea, 50 mM Tris (pH 8), 75 mM NaCl, and a cocktail of protease inhibitors (Roche Complete-mini EDTA-free). After scraping, cells were subjected to 2 × 30 s of probe sonication, 20 min of incubation on ice, followed by 10 min of centrifugation at 21,000 × g and 4 °C. The protein content of the supernatant was estimated using BCA. The proteins were reduced with 5 mM dithiothreitol for 30 min at 55 °C, alkylated with 10 mM iodoacetamide in the dark for 30 min at room temperature, and quenched with an additional 5 mM dithiothreitol for 15 min at room temperature. The proteins were diluted to 1.8 M urea and then digested with sequencing grade trypsin (Pierce) at a 1:50 enzyme to substrate ratio for 12 h at 37 °C. The digestion was quenched by adding 10% trifluoroacetic acid to achieve approximately pH 2. Resulting peptides were desalted with 100 mg tC18 SepPak cartridges (Waters) using vendor-provided protocols and dried with vacuum centrifugation. Peptides were brought to 1 µg/3 µl in 0.1% formic acid (buffer A) prior to mass spectrometry acquisition. For the reproducibility experiments and to build a chromatogram library we pooled aliquots from all six time points for three of the replicates to ensure that the pool contained virtually every peptide present in the individual time points.

With the phosphoproteomics experiment, four replicates were performed for each of the four conditions: 20 min EGF (100 ng/ml) or PBS stimulation following 4 h starvation, and 20 min EGF/PBS stimulation following 16 h starvation. Sample generation and processing was performed in the same fashion with the following exceptions: (1) in addition to protease inhibitors, a cocktail of phosphatase inhibitors (50 mM NaF, 50 mM β-glycerophosphate, 10 mM pyrophosphate, and 1 mM orthovanadate) was also added to the lysis buffer, (2) proteins were digested for 14 h, and 3) phosphopeptides were enriched using immobilized metal affinity chromatography (IMAC) using Fe-NTA magnetic agarose beads (Cube Biotech). Enrichment was performed with a KingFisher Flex robot (Thermo Scientific), which incubated peptides with 150 µl 5% bead slurry in 80% acetonitrile, 0.1% TFA for 30 min, washed them three times with the same solution, and eluted them with 60 µl 50% acetonitrile:1% NH4OH. Phosphopeptides were then acidified with 10% formic acid and dried. Phosphopeptides were brought to 1 µg/3 µl in 0.1% formic acid assuming a 1:100 reduction in peptide abundance from the IMAC enrichment. Again, to build a chromatogram library we pooled aliquots from all four conditions for three of the replicates to ensure that the pool contained virtually every peptide present in the individual conditions.

**Yeast cell culture and sample preparation.** Yeast strain BY4741 (Dharmacon) was cultured at 30 °C in YEPD and harvested at mid-log phase. Cell pellets were lysed in a buffer of 8 M urea, 50 mM Tris (pH 8), 75 mM NaCl, 1 mM EDTA (pH 8) using 7 cycles of 4 min bead beating with glass beads followed by one minute rest on ice. Lysate was collected by piercing the tube, placing it into an empty eppendorf, and centrifuging for 1 min at 3000 × g and 4 C. Insoluble material was removed from the lysate by 15 min centrifugation at 21,000 × g and 4 C. The protein content of the supernatant was estimated using BCA. The proteins were reduced with 5 mM dithiothreitol for 30 min at 55 °C and alkylated with 10 mM iodoacetamide in the dark for 30 min at room temperature. The proteins were diluted to 1.8 M urea and then digested with sequencing grade trypsin (Pierce) at a 1:50 enzyme to substrate ratio for 16 h at 37 °C. The digestion was quenched using 5 N HCl to achieve approximately pH 2. Resulting peptides were desalted with 30 mg MCX cartridges (Waters) and dried with vacuum centrifugation. Peptides were brought to 1 µg / 3 µl in 0.1% formic acid (buffer A) prior to mass spectrometry acquisition.

**Mixtures of yeast and HeLa cells.** Mixtures of digested yeast and HeLa peptides were combined in the following yeast:HeLa ratios: 1:0, 0.7:0.3, 0.5:0.5, 0.3:0.7, 0.1:0.9, 0.07:0.93, and 0:1, where concentrations were assumed from protein-level BCA analyses. Ratio mixing bias (caused by bias in BCA estimates from assuming Bovine serum albumin as a standard) were determined by regression across all ratios (both yeast:HeLa and HeLa:yeast) using a linear model using the expected ratio of the measured species as a regression term. After correction, the recalculated ratios were determined to be 1:0, 0.78:22, 0.56:0.44, 0.34:0.66, 0.11:0.89, 0.078:0.922, and 0:1.

**LC mass spectrometry.** Peptides were separated with a Waters NanoAcquity UPLC and emitted into a Thermo Q-Exactive HF tandem mass spectrometer. Pulled tip columns were created from 75 µm inner diameter fused silica capillary in-house using a laser pulling device and packed with 3 µm ReproSil-Pur C18 beads

(Dr. Maisch) to 300 mm. Trap columns were created from 150 µm inner diameter fused silica capillary fritted with Kasil on one end and packed with the same C18 beads to 25 mm. Solvent A was 0.1% formic acid in water, while solvent B was 0.1% formic acid in 98% acetonitrile. For each injection, 3 µl (approximately 1 µg) was loaded and eluted using a 90-minute gradient from 5 to 35% B, followed by a 40 min washing gradient. Data were acquired using either data-dependent acquisition (DDA) or data-independent acquisition (DIA). Three DDA and DIA HeLa and yeast technical replicates were acquired by alternating between acquisition modes to minimize bias. Serum-starved HeLa acquisition was randomized within blocks to enable downstream statistical analysis.

**DDA acquisition and processing.** The Thermo Q-Exactive HF was set to positive mode in a top-20 configuration. Precursor spectra (400–1600 m/z) were collected at 60,000 resolution to hit an AGC target of 3e6. The maximum inject time was set to 100 ms. Fragment spectra were collected at 15,000 resolution to hit an AGC target of 1e5 with a maximum inject time of 25 ms. The isolation width was set to 1.6 m/z with a normalized collision energy of 27. Only precursors charged between +2 and +4 that achieved a minimum AGC of 5e3 were acquired. Dynamic exclusion was set to "auto" and to exclude all isotopes in a cluster. Thermo RAW files were converted to mzXML format using ReAdW and searched against a Uniprot Human FASTA database (87613 entries) with Comet (version 2015.02v2), allowing for variable methionine oxidation, and n-terminal acetylation. Cysteines were assumed to be fully carbamidomethylated. Searches were performed using a 50 ppm precursor tolerance and a 0.02 Da fragment tolerance using fully tryptic specificity (KR|P) permitting up to two missed cleavages. Search results were filtered to a 1% peptide-level FDR using Percolator (version 3.1).

**DIA acquisition and processing.** For each chromatogram library, the Thermo Q-Exactive HF was configured to acquire six chromatogram library acquisitions with 4 m/z DIA spectra (4 m/z precursor isolation windows at 30,000 resolution, AGC target 1e6, maximum inject time 55 ms) using an overlapping window pattern from narrow mass ranges using window placements optimized by Skyline (i.e., 396.43–502.48, 496.48–602.52, 596.52–702.57, 696.57–802.61, 796.61–902.66, and 896.6–1002.70 m/z). See Supplementary Figure 10 and Supplementary Data 2 for the actual windowing scheme. Two precursor spectra, a wide spectrum (400–1600 m/z at 60,000 resolution) and a narrow spectrum matching the range (i.e., 390–510, 490–610, 590–710, 690–810, 790–910, and 890–1010 m/z) using an AGC target of 3e6 and a maximum inject time of 100 ms were interspersed every 18 MS/MS spectra.

For quantitative samples, the Thermo Q-Exactive HF was configured to acquire 25 × 24 m/z DIA spectra (24 m/z precursor isolation windows at 30,000 resolution, AGC target 1e6, maximum inject time 55 ms) using an overlapping window pattern from 388.43 to 1012.70 m/z using window placements optimized by Skyline. See Supplementary Figure 11 and Supplementary Data 2 for the actual windowing scheme. Precursor spectra (385–1015 m/z at 30,000 resolution, AGC target 3e6, maximum inject time 100 ms) were interspersed every 10 MS/MS spectra. Phosphopeptide samples were analyzed in the same way using 20 × 20 m/z DIA spectra in an overlapping window pattern from 490.47 to 910.66 m/z.

All DIA spectra were programed with a normalized collision energy of 27 and an assumed charge state of +2. Thermo RAW files were converted to mzML format using the ProteoWizard package (version 3.0.7303) where they were peak picked using vendor libraries. A HeLa-specific Bibliospec[20] HCD spectrum library was created from Thermo Q-Exactive DDA data using Skyline (version 3.1.0.7382). This library is comprised of 39 SCX and high-pH reverse phase fractionated raw files using multiple HPLC gradients to vary the local peptide matrix. This BLIB library and accompanying iRTDB normalized retention time database were converted into a ELIB library and used to search the mzMLs for peptides. EncyclopeDIA searches DIA data using +1H and +2H b/y ion fragments that could be found in library spectra. EncyclopeDIA was configured with default settings (10 ppm precursor, fragment, and library tolerances, considering both B and Y ions, and trypsin digestion was assumed). EncyclopeDIA was configured to use Percolator version 3.1. Phosphopeptides were processed the same way except a HeLa-specific phosphopeptide HCD spectrum library was used[38] and phosphopeptides detected in EncyclopeDIA searches were localized using Thesaurus[39].

Further validation of the HeLa replicate dataset was performed using Skyline-daily version 4.1.1.18151. Precursors were filtered between the isolated m/z range of 388.4 to 1000.7 with a minimum of 6 measurable fragment y-ion and b-ion (charge 1 or 2) between 300 and 2000 m/z, not including y1, y2, b1, or b2. The fragment ions with the six most intense peaks from the libraries within these limits were chosen along with the first three precursor isotopes to be extracted from MS1, both set to extract within 10 ppm mass error from the centroided (and demultiplexed for MS/MS) spectra. Two iRT libraries were built (for the HeLa-specific DDA library and the HeLa-specific chromatogram library, respectively) using 73 reliably detected peptides were chosen as iRT library anchors across the retention time range. Chromatogram extraction was set to apply to all spectra within 10 min of predicted retention times using these iRT libraries. A mProphet[40] model was trained using the target/decoy strategy (with the "Retention time difference squared" excluded) and applied without any run-to-run alignment. Please see Supplementary Note 2 for further details.

**Overlapping DIA deconvolution**. When using the overlapping DIA scheme, every spectrum in the entire raw file must be deconvoluted. In an effort to maintain consistency between analysis techniques, we used MSConvert to deconvolute RAW files in this study. However, we have also implemented a simple deconvolution algorithm in EncyclopeDIA that can be performed on-the-fly while reading spectra in a narrow I/O buffer. In a DIA data set, at each cycle ($T$) every MS/MS spectrum ($S_{Ti}$) comprises fragments from precursors within the precursor isolation window ($i$). Spectra in consecutive half cycles are overlapped by 50%, such that precursors from the lower 50% of the window in MS/MS spectrum $S_{Ti}$ should also be present in the previous/next half cycles lower offset spectra ($S_{(T-1)(i-1)}$ and $S_{(T+1)(i-1)}$) while precursors from the upper 50% of the window should also be present in the corresponding upper offset spectra ($S_{(T-1)(i+1)}$ and $S_{(T+1)(i+1)}$). We divide these windows into two bins and attempt to determine which fragments were derived from precursors in the upper half or the lower half using previous and next half cycles. Fragment ions that are found exclusively on the lower previous/next spectra ($S_{(T-1)(i-1)}$ and $S_{(T+1)(i-1)}$) are assigned to the lower bin, while those found exclusively in the upper previous/next spectra ($S_{(T-1)(i+1)}$ and $S_{(T+1)(i+1)}$) are assigned to the upper bin. Ions that are found in both sets of spectra are assigned proportionally to each bin where the proportion is set to the summed peak intensity for both spectra, e.g.: $(S_{(T-1)(i-1)} + S_{(T+1)(i-1)}) / (S_{(T-1)(i-1)} + S_{(T+1)(i-1)} + S_{(T-1)(i+1)} + S_{(T+1)(i+1)})$ for the lower bin. Peaks that are found in none of the previous and next overlapping spectra are assumed to be noise. New spectra are built from the deconvoluted peaks in both the lower and upper bins. Since this algorithm only needs to consider three half cycles at a time, deconvolution can happen quickly and in memory, with minimal impact on file reading speeds.

**Decoy library entries**. A decoy library entry is created for every target library entry. To generate a decoy, first the target peptide sequence (except for digestion enzyme-specific termini) is reversed, ensuring that the decoy maintains its appearance as a tryptic peptide. Then fragment ions corresponding to amino acids (B/Y for CID, C/Z/Z + 1 for ETD) or their expected neutral losses due to modifications (e.g. phosphorylation) are calculated for both target and decoy entries. If the precursor charge state is greater than +2, then +2 fragment ions are also considered. Uncommon neutral loss ions such as A-type ions or loss of water or ammonia are not considered to limit the likelihood of false detections. Fragment ions that correspond to target sequence m/zs are transferred to new decoy m/zs such that their ion type and index are kept consistent. Delta mass errors in each fragment ion are also maintained to preserve consistency, and all peaks corresponding to the fragment delta mass window are transferred if the library is collected in profile mode. Ions that cannot be assigned to amino acids (such as those corresponding to precursor ions, background noise or interference) are not used by EncyclopeDIA.

**Ion weighting estimation**. While searching, a unique background is calculated for each precursor isolation window using the prevalence of each fragment ion in the library spectra considered for that window (Supplementary Figure 12). This background helps estimate the interference frequency for any given ion and is used to weight some scores. This distribution is calculated as the frequency that any nominal m/z fragment ion (rounded by truncation) appears in entries from the library within the specified precursor window filter. m/z frequencies are calculated out to 4000 and a pseudocount is applied to every m/z bin to avoid divide by zero frequency errors.

**Primary scoring and feature scoring functions**. The primary score in EncyclopeDIA conceptually draws on the X!Tandem HyperScore. Unlike scoring functions like XCorr in Sequest, the HyperScore does not attempt to account or penalize for ions that do not match the peptide in question, making it ideal for DIA analysis where coeluting peptides are common. The score function is the weighted dot product of the intensities in the acquired spectrum ($I$) and the library spectrum ($P$), weighted by a correlation score vector ($C$), which is discussed in detail in the Chromatogram Library ELIB Generation section. Again, any ions in the library spectrum that do not correspond to the amino acid sequence are not considered in this score. The dot product is multiplied by the factorial of the number of matching ions:

$$\text{Primary score} = \log_{10}\left(\left(\sum_{i=0}^{n} I_i \cdot P_i \cdot C_i\right) \cdot n!\right) \quad (1)$$

Sometimes modified peptides (for example, oxidized peptides) are present in the same precursor isolation window as their unmodified forms. Since often these peptides share several fragment ions in common, we require that at least 25% of the score contribution for modified peptides come from ions that exclusively indicate that modification in cases where any of up to four isotopic peaks from the modified/unmodified peptide pairs fall in the same window.

Several more computationally expensive secondary feature scores (Supplementary Data 3) are calculated once peaks are assigned. Briefly, the scores are divided to cover various classes of features: overall scoring (deltaCN, eValue, logDotProduct, logWeightedDotProduct, xCorrLib, xCorrModel), fragment ion

accuracy (sumOfSquaredErrors, weightedSumOfSquaredErrors, numberOfMatchingPeaks, averageAbsFragDeltaMass, averageFragmentDeltaMass), precursor ion accuracy (isotopeDotProduct, averageAbsPPM, averagePPM), and retention time accuracy (deltaRT). The deltaRT score is only used after retention time alignment has been performed. All of these scores are fed to Percolator 3.1 for target/decoy FDR analysis.

**Retention time alignment**. Accuracy and stability of retention time alignments is critical for EncyclopeDIA. Consequently, we designed an algorithm that works analogous to how we visualize densities. This approach uses two-dimensional kernel density estimates (KDE) that are much less prone to failure as compared to typical line fitting approaches such as LOESS in situations with grossly variable numbers of points and outliers. In this approach each X/Y coordinate is estimated as a symmetrical, two-dimensional kernel based on a cosine-based Gaussian approximation. Following Silverman's rule[41] the KDE bandwidth is set to:

$$\text{Bandwidth} = N^{-\frac{1}{6}} \cdot \left(\frac{\text{stdev}(x) + \text{stdev}(y)}{2}\right) \quad (2)$$

where $N$ is the number of matched peptides. The kernel's standard deviation is set to the bandwidth (analogous to full width at half max) divided by $2\sqrt{2 \cdot \ln(2)}$. This distribution is stamped at every X/Y coordinate on a 1000 by 1000 grid mapping from the lowest and highest retention times in both the X and Y dimensions. Once the KDE is calculated, the optimal fit is traced using a ridge walking algorithm that traces the mode of the KDE across retention time (Supplementary Figure 13). In this algorithm the highest point in the KDE is identified and the line is fit in increasing retention time by moving to the highest local grid point to the north (increased sample retention time), east (increased library retention time), or northeast. If north and east are both the highest local point, then the line moves to the northeast. This is performed iteratively until the line is fit across the increasing retention time. Then the same ridge walk is performed in decreasing retention time by moving south, west, or southwest. This approach forces a monotonic line (it can never find a negative retention time change) that follows where the most number of X/Y coordinates lie.

**Retention time alignment mixture model**. After the alignment is performed, we use the delta retention time data to produce a mixture model to determine outliers. We calculate a Gaussian distribution representing correct retention time matches using the median delta retention time as the Gaussian mean and interquartile range divided by 1.35 as the Gaussian standard deviation. We use a unit distribution to represent incorrect retention time matches. Starting where the distribution priors are set to 0.5, we run 10 iterations of a PeptideProphet-like mixture model[42] to fit the two distributions to the delta retention time data using an Expectation Maximization algorithm[43]. Peptide matches with posterior error probability estimations that are less than 5% likely to be in the correct retention time distribution are considered outliers.

**Retention time alignment across experiments**. For each passing peptide, we determine the experiment that produced the best scoring match and set that match aside as a canonical peptide representation. We chose the experiment with the most canonical peptides as an anchor and retention time align all of the experiments (and their canonical peptides) to that anchor. Mixture models (described above) for these retention time alignments are calculated and outliers are removed if the local-anchor delta retention time is less than 0.1% likely to fit the mixture model. New retention times for outlier-removed peptides and peptides that were only assigned globally are inferred using the anchor retention time.

**Peptide and protein FDR filtering across experiments**. We concatenate peptide feature files from all experiments in a study and run Percolator 3.1 to perform global peptide FDR filtering at 1%. Using this list of peptides, we generate a parsimonious list of protein groups using a greedy algorithm. Here peptides are assigned to protein groups with the highest protein score:

$$\text{Protein score}(P) = N - \sum_{p \in P}^{N} (\text{PEP}_p) \quad (3)$$

where the the sum of the peptide ($p$) posterior error probabilities ($\text{PEP}_p$) is subtracted from the number of peptides ($N$) assigned to that protein ($P$). Protein groups are sorted on the lowest $\text{PEP}_p$ assigned to them[18] and then stringently target/decoy filtered to 1% protein FDR.

**Automated transition refinement**. Fragment ion interference is common when analyzing wide-window data. While fragment ions that show interference may still be useful for detecting peptides, those ions must be screened prior to quantitation to ensure an accurate measurement. We designed a non-parametric approach to selecting the best ions for quantitation. We first Savizky-Golay smooth[44] the fragment ion chromatograms and then normalize them to have unit integrated intensity. To simplify the smoothing mathematics, we make the assumption that cycle times are consistent within the time frame of a single peak, thus removing the

need for interpolation over retention time. After normalization the chromatograms of quantitatively useful ions line up while those of interfered ions will have either higher or lower unit-normalized intensities at different retention times. We calculate the median normalized intensity at each retention time point as an approximation for the peptide peak shape. We then determine peak boundaries by tracing descent of the median peak shape from the maximum normalized intensity on either side of the peak. The boundaries are set to the minimum point at which the median peak trace starts increasing for >2 consecutive spectra or any point where the trace drops to less than 1% of the maximum. At that point we calculate a Pearson's correlation coefficient for the similarity between each fragment ion chromatogram with that of the median peak shape between those boundaries. Peaks that match with a correlation coefficient of at least 0.9 are considered quantitative, while those that match with coefficients of at least 0.75 are considered useful for detection purposes.

**Fragment ion quantification and background subtraction**. We calculate trapezoidal peak areas across Savitsky-Golay smoothed chromatograms. Analogous to Skyline, peak intensities are background subtracted by removing a peak area rectangle with a height equal to the largest intensity of either of the boundary edges. If the area of the rectangle is larger than the area of the peak the intensity is set to zero.

**Peptide quantification and transition choice across experiments**. Transition interference changes on a sample by sample basis. We rank quantitative transitions (>0.9 correlation) based on the sum of their correlation scores across all experiments (effectively counting the number of samples in which they are observed). In addition, for each transition we calculate a global interference score:

$$\text{Interference score}(t) = \frac{\sum_s I_{t,s}[C_{t,s} < 0.9]}{\sum_s I_{t,s}[C_{t,s} \geq 0.9]} \tag{4}$$

which represents the sum of transition ($t$) intensities ($I_{t,s}$) across experiments ($s$) that show interference ($C_{t,s} < 0.9$) over those that do not ($C_{t,s} \geq 0.9$). Transitions with interference scores > 0.2 are deemed untrustworthy for quantification and are dropped. Peptide quantities are set to the sum of the top five transitions that pass these criteria, where peptides with fewer than 3 quantitative transitions are not carried forward. We require additional stringent criteria for our time course study. Specifically, we required that each peptide be measured in every replicate of at least one time point, and that cross experiment CVs (estimated using quantities from each time point corrected with a linear model) be less than 20%.

**Protein quantification and statistical testing**. Protein quantities were calculated as the sum of peptide quantities. We used Extraction of Differential Gene Expression (EDGE) 3.6[36] to statistically test for reproducible changes across the time course study. We performed k-means clustering of proteins that passed an EDGE q-value < 0.01 using five groups using 1000 random starting points with 1000 iterations. We estimated five groups by calculating the sum of within squared errors of each K model from 1 to 15 and estimating the first point where the change in the sum of within squared errors was flat (Supplementary Figure 14).

**Gene Ontology enrichment**. We performed Gene Ontology enrichment of significantly changing proteins using the online PANTHER Overrepresentation Test[45] (release 20170413) with the Homo sapiens Gene Ontology database (release 2017-10-24) using a background of all proteins consistently detected in our experiments. After removing terms with fewer than 20 proteins (to avoid weakly powered classes) and more than 1000 proteins (to avoid vague classes), we applied Benjamini-Hochberg FDR correction and filtered enrichment tests to a FDR < 0.05.

**Code availability**. EncyclopeDIA is implemented in Java 1.8 as both a command line and a stand-alone GUI application. EncyclopeDIA supports the HUPO PSI mzML standard for reading raw MS/MS data, and can construct DLIB DDA-based spectrum libraries from Skyline/Bibliospec BLIB files, NIST MSP files, or HUPO PSI TraML files. Additionally, EncyclopeDIA results can be imported into Skyline[25] to enable further visualization and downstream processing. EncyclopeDIA is heavily optimized and multi-threaded such that searches can be performed on conventional desktop computers with limited RAM and processing power. We have released source code and cross platform (Windows, Mac OS X, Linux) binaries for EncyclopeDIA on Bitbucket at: https://bitbucket.org/searleb/encyclopedia under the open source Apache 2 license.

**Data availability**
All mass spectrometry mzML and RAW data files (see Supplementary Data 4 for raw data annotations) are available on the Chorus Project (project identifier 1433, chromatogram library data for human [https://chorusproject.org/anonymous/download/experiment/32fa43c0f9ba486eb3eedeb689f87765] and yeast [https://chorusproject.org/anonymous/download/experiment/b98531fe7fe246cbb7e45ce065fe54a9], serum starvation data proteomics [https://chorusproject.org/anonymous/download/experiment/e0659292e919414787ec112dca4c57c1] and phosphoproteomics [https://chorusproject.

org/anonymous/download/experiment/c24893cd7115446dab4d7eeb7fde2506] data) and at the MassIVE proteomics repository (project identifier MSV000082805 [https://massive.ucsd.edu/ProteoSAFe/dataset.jsp?task = e340c79fbdc64e14a710265761bfeed5]). All other data supporting the findings of this studz are available from the corresponding author on reasonable request. A reporting summary for this article is available as a Supplementary Information file.

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

## Acknowledgements

We would like to thank members of the Villén and MacCoss labs for critical discussions. We additionally thank N. Shulman for implementing Skyline visualization of EncyclopeDIA reports, and S. Just, P. Seitzer, and S. Ludwigsen for EncyclopeDIA bug reports and patches. B.C.S. is supported by F31 GM119273; L.K.P. is supported by F31 AG055257. This work is supported by P41 GM103533, R21 CA192983, and U54 HG008097 to M.J.M.; and R35 GM119536, R01 AG056359, and a research grant from the W.M. Keck Foundation to J.V.

## Author contributions

B.C.S. and M.J.M. conceived the study. B.C.S., R.T.L., and M.J.M designed the experiments. B.C.S., L.K.P., and R.T.L. performed the experiments. B.C.S. designed and wrote the software with input from L.K.P., J.D.E., and Y.S.T.. B.C.S. and B.X.M. analyzed the data. M.J.M. and J.V. supervised the work. B.C.S., L.K.P., J.D.E., Y.S.T., R.T.L., B.X.M., J. V., and M.J.M. wrote the paper.

## Additional information

**Competing interests:** The MacCoss Lab at the University of Washington (members B.C.S., L.K.P., J.D.E., Y.S.T., B.X.M. and M.J.M.) has a sponsored research agreement with Thermo Fisher Scientific, the manufacturer of the instrumentation used in this research. Additionally, M.J.M. is a paid consultant for Thermo Fisher Scientific. The remaining authors declare no competing interests.

