## [Peer Review File · Nature Communications]

Reviewers' Comments:

Reviewer #1:

Remarks to the Author:

Searle et al. present an experimental workflow for the construction of chromatogram libraries that store fragment ion chromatographic peak shapes, which allow for sensitive peptide detection in samples acquired for quantification. While the approach is interesting and likely deserves publication, the manuscript in the current form has significant shortcomings that the authors have to fix.

1) The validation of the new method is poor. The authors just show that they can determine some global proteomic changes in a dataset. This does not prove anything about the quantification accuracy of the new method. The authors have to use a controlled benchmark dataset in which the ground truth is known to prove that their new method is superior to current state of the art. Suitable benchmark datasets can be generated by mixing proteomes from different species in suitable ratios or by spiking protein standards into a 1:1 background of complex proteome, e.g. UPS1 vs. UPS2 proteomic standards from Sigma-Aldrich/Merck.

2) On p.3 the authors claim 'in DDA workflows each individual sample is informatically processed alone to account for stochastic variation in data acquisition'. This is not true. Modern DDA data analysis platforms analyze all quantitative samples together and match MS1 features across runs for quantification. For instance, the MaxQuant platform has the 'Matching between runs' feature which does exactly this. The authors are describing the state of DDA proteomics as it was ten years ago.

3) Relating to 2) it is doubtful if the authors use in their comparison the best possible way of analyzing DDA data. Just doing a Comet search does not seem to be state of the art. Comparing to MaxQuant with and without 'Match between runs' would be much more informative to the reader, since that is what the majority of proteomics labs are doing.

Reviewer #2:

Remarks to the Author:

In their manuscript entitled „ Comprehensive peptide quantification for data independent acquisition mass spectrometry using chromatogram libraries “ Searle et al. describe a novel DIA approach based on chromatogram libraries, which capture fragment ion chromatographic peak shape and retention time for all detectable peptides in an experiment. To optimally evaluate the dataset, the authors also present EncyclopeDIA, a software tool for generating and searching chromatogram libraries. Using their approach, the authors are able to quantify about 50% more peptides compared to a spectral-library based approach. Finally, they demonstrate the performance of the novel workflow by quantifying proteins in human and yeast cells. The manuscript is excellently written and methods and results are clearly described. The data are of high quality. The authors present a novel, chromatogram library based workflow enabling DIA-based deep coverage quantitative proteomics experiments.

I have only a few points that the authors should address:

Q1) Why was the mass range limited to $m/z = 400 - 1000$? I recommend that the authors investigate the performance of larger mass ranges (e.g. $350 - 1100$, using 2.5 Da windows), or $350 - 1250$, using 3 Da windows. This alternative acquisition strategies might improve proteome coverage, but will obviously slightly decrease the specificity of the library generation.

Q2) How does the workflow perform for more complex samples? Did the authors investigate the quantitative performance (precision/accuracy of reported ratios) of their workflow using mixed proteome samples (as described by Navarro et al, PMID: 27701404)?

Q3) How does the performance of the new method compare to the recently described strategy by Bruderer et al. (PMID: 29070702)

Reviewer #3:

Remarks to the Author:

In their manuscript "Comprehensive peptide quantification for data independent acquisition mass spectrometry using chromatogram libraries", Searle et al. present a new workflow for identification and quantification of DIA data based on chromatogram libraries. DIA data has gained increasing traction in the last few years especially through the development of SWATH-MS and the development of new tools and approaches in DIA data analysis is important in the field. The proposed study is an interesting contribution to the field as it combines multiple approaches previously described into a new workflow and describes a open-source tool to run the data analysis. The novelty of the method lies in a novel way of constructing spectral libraries for peptide-centric DIA analysis and incremental improvements to the already published approaches for peptide-centric analysis.

While the development of novel methods for DIA data analysis is of high interest, the current manuscript lacks in several major aspects. First and foremost, it is unclear whether the major advance described here is a new tool or a new way to construct spectral libraries. While the development of a "chromatographic library" seems to be the main advance, the authors remain surprisingly vague about what exactly their "chromatographic library" really contains and how it differs from traditional assay libraries. It is also not clear whether the new "chromatographic libraries" are actually performing better than traditional assay libraries since the authors fail to make an appropriate comparison. Another major question regarding the new library generation method is whether the authors employ appropriate control of the false discovery rate when constructing the library. Importantly, the authors should put their work in proper context with previous work, citing the appropriate papers where they draw their inspiration for their improvements from, they discuss and cite a lot of work from the MacCoss lab but fail to mention other relevant work. Finally, the authors produce a new tool for DIA data analysis called EncyclopeDIA which uses the new assay libraries, but fail to compare it to existing approaches.

Major comments:

- The authors propose a new method for identifying peptides and generating spectral libraries directly from DIA data. They report a fantastic increase of almost 2-fold in the number of peptides in their narrow-window DIA identification compared to their previous tool using Walnut / PECAN. To achieve this, the authors used a spectral library search, however the authors do not adequately describe how they achieved spectral library search in DIA data and how they perform error control in DIA datasets. The authors should (i) describe their approach in detail and (ii) report how many identifications tools built for spectral library searches in DIA data (such as MSPLIT-DIA) report. Since all of their subsequent results rest on their initial spectral library with 99.6k unique peptides, I suggest to better describe the library generation procedure and validate their findings of 99.6k peptides with MSPLIT-DIA or SpectraST.
- The authors should cite appropriate literature describing similar advances in the field and acknowledge previous achievements in the field. For example, they are not the first ones to report better identification and quantification in DIA compared to DDA (this has been consistently reported since the first high throughput peptide-centric DIA papers were published). Also their approach of RT normalization seems similar to the high-precision iRT approach by Bruderer et al (Proteomics, 2016). The authors should also cite and contrast their work on fragment ion interference removal to recent work such as mapDIA and SWATHProphet that essentially aim to achieve something similar.
- It is currently unclear what the "chromatographic library" generated here really entails and what

its novel aspect is. Currently the main advance seems to be the fact that the retention times in the library are more accurate than in previous, iRT based libraries since they are derived on the same column. Also the authors speculate that the fragment ion intensities may be more accurate. However, the authors need to clarify how their approach differs from the high-precision iRT concept (Bruderer et al, Proteomics, 2016) and whether their "chromatographic library" features any additional improvements compared to the "high precision iRT library" described in this publication.

- The authors fail to compare their new tool with existing tools. The authors have access to the Skyline tool developed in the same lab which has recently shown to perform equivalent to other tools (OpenSWATH, Spectronaut, SCIEX etc. see Navarro et al 2016). Without proper benchmarking and tool comparison, the readers are left in the dark whether the new tool / workflow provides a substantial improvement over existing approaches. It seems straight forward for example to load the 99.6k (or a subset thereof) peptide library into either Skyline, OpenSWATH or Spectronaut and perform peptide-centric analysis. This direct comparison would quantify the advantage that the new "chromatographic libraries" approach brings in conjunction with existing tools like Skyline.

- I seem to not find a comparison of the new library generation approach to the traditional library generation approach. Maybe I missed it and the authors can point me to the appropriate figure, but it seems straight forward to use a large scale library such as the 166.4k peptide library the authors have access to and directly search the wide window DIA data. This would showcase whether the somewhat labour-extensive step of intermediately creating narrow window DIA data to create a 2nd library is necessary and improves over the straight forward approach of directly using the initial 166.4k peptide library for peptide-centric extraction. If this is what Figure 2a shows, please be more clear about this.

- The authors claim that their fragment ion interference removal provides improved quantitative accuracy. However, they do not provide evidence for such a claim. Either they need to remove the claim or show data to back this up - e.g. show improved CV / quantitative accuracy when turning on their interference removal.

- In addition to citing PECAN as a library-free tool the authors should also cite Spectronaut and DIA-Umpire.

- It is not clear how the authors calculate "global peptide FDR across all experiments" - did they use the method described in ref 18?

- The authors should describe more of their experimental setup in the main paper, e.g. gradient length and instrument used when describing the number of identified peptides -- or number of samples and biological / technical replicates used when describing their biological study.

Minor comments:

- typo in decoy library method: "insuring" should be "ensuring"

Reviewer #1 (Remarks to the Author):

Searle et al. present an experimental workflow for the construction of chromatogram libraries that store fragment ion chromatographic peak shapes, which allow for sensitive peptide detection in samples acquired for quantification. While the approach is interesting and likely deserves publication, the manuscript in the current form has significant shortcomings that the authors have to fix.

1) The validation of the new method is poor. The authors just show that they can determine some global proteomic changes in a dataset. This does not prove anything about the quantification accuracy of the new method. The authors have to use a controlled benchmark dataset in which the ground truth is known to prove that their new method is superior to current state of the art. Suitable benchmark datasets can be generated by mixing proteomes from different species in suitable ratios or by spiking protein standards into a 1:1 background of complex proteome, e.g. UPS1 vs. UPS2 proteomic standards from Sigma-Aldrich/Merck.

As the reviewer suggested, we have now performed a quantitative benchmarking experiment using mixed proteomes of yeast diluted in a HeLa background. We performed five dilutions in triplicate and measured protein intensities relative to 100% yeast:

In general, quantitative measurements made by EncyclopeDIA agree closely with the expected ratios. We agree that this was an important validation and we now present this result as a new Figure 5.

2) On p.3 the authors claim 'in DDA workflows each individual sample is informatically processed alone to account for stochastic variation in data acquisition'. This is not true. Modern DDA data analysis platforms analyze all quantitative samples together and match MS1 features across runs for quantification. For instance, the MaxQuant platform has the 'Matching between runs' feature which does exactly this. The authors are describing the state of DDA proteomics as it was ten years ago.

We have removed this sentence.

3) Relating to 2) it is doubtful if the authors use in their comparison the best possible way of analyzing DDA data. Just doing a Comet search does not seem to be state of the art. Comparing to MaxQuant with and without 'Match between runs' would be much more informative to the reader, since that is what the majority of proteomics labs are doing.

The primary goal of our DDA to DIA comparison (Figures 2 and 3) was to capture the run-to-run variability in detections. As MaxQuant can map MS/MS identifications to MS1 features that have never been sampled in other runs with the "match between runs" feature, EncyclopeDIA can also map MS2 detected peptides in DIA to other MS2 signals in other runs using run-to-run alignment. That said, this comparison was not the objective of this experiment. A major goal of proteomics is to comprehensively detect peptides in individual runs without being able to computationally align between technical replicates, and we were trying to demonstrate the percentage of the detectable proteome that could be seen in each replicate individually. As such, replicate analyses for both DDA and DIA were performed independently for the experiments presented in Figures 2 and 3, and no run-to-run alignment was employed in either analysis strategy.

While Comet is a fork of the original SEQUEST scoring algorithm published originally 24 years ago, we believe that when coupled with Percolator it remains state of the art. The Comet codebase has been continuously improved and maintained since 1994, making it arguably the most mature and widespread search engine in the proteomics community. Furthermore, while we appreciate the capabilities of MaxQuant, its source code remains unavailable. While the value of MaxQuant is undeniable, two editorials in Nature Methods make it clear that the release of code is an important component to transparency and reproducible science (doi:10.1038/nmeth0307-189; doi:10.1038/nmeth.2880).

Reviewer #2 (Remarks to the Author):

In their manuscript entitled „ Comprehensive peptide quantification for data independent acquisition mass spectrometry using chromatogram libraries “ Searle et al. describe a novel DIA approach based on chromatogram libraries, which capture fragment ion chromatographic peak shape and retention time for all detectable peptides in an experiment. To optimally evaluate the dataset, the authors also present EncyclopeDIA, a software tool for generating and searching chromatogram libraries. Using their approach, the authors are able to quantify about 50% more peptides compared to a spectral-library based approach. Finally, they demonstrate the performance of the novel workflow by quantifying proteins in human and yeast cells. The manuscript is excellently written and methods and results are clearly described. The data are of high quality. The authors present a novel, chromatogram library based workflow enabling DIA-based deep coverage quantitative proteomics experiments.

I have only a few points that the authors should address:

Q1) Why was the mass range limited to $m/z = 400 - 1000$? I recommend that the authors investigate the performance of larger mass ranges (e.g. $350 - 1100$, using 2.5 Da windows), or $350 - 1250$, using 3 Da windows. This alternative acquisition strategies might improve proteome coverage, but will obviously slightly decrease the specificity of the library generation.

Because increasing the m/z range of the chromatogram library generation step only requires an additional run or two, we have explored collecting eight overlapped window gas-phase fractionated injections (from $400 - 1200$ m/z with 4 m/z windows) rather than just six (from $400 - 1000$ m/z with 4 m/z windows). We find that increasing the number of fractions does indeed increase the library size (panel a). However, the wide-window DIA experiments need to be similarly enlarged in order to benefit from this larger library. We ran triplicate experiments to assess this effect using the following overlapped window data acquisition schemes: $500 - 900$ m/z with 25×16 m/z windows, $400 - 900$ m/z with 25×20 m/z windows, $400 - 1000$ m/z with 25×24 m/z windows, $400 - 1100$ m/z with 25×28 m/z windows, and $400 - 1200$ m/z with 25×32 m/z windows. Here window width scaled to maintain equivalent cycle times. We found that while m/z range did not substantially affect results when searching with a spectrum library, when searching with a chromatogram library we saw the highest number of peptides detected using $400 - 1000$ m/z wide-window searches (panel b). This result confirmed our intuition that widening isolation windows beyond 24 m/z can have detrimental results. These results are summarized below:

We have not added this experiment (or figure) into the manuscript but are happy to do so if the reviewer deems it useful.

Q2) How does the workflow perform for more complex samples? Did the authors investigate the quantitative performance (precision/accuracy of reported ratios) of their workflow using mixed proteome samples (as described by Navarro et al, PMID: 27701404)?

As discussed above in our response to Reviewer 1, we have now performed a quantitative experiment using mixed proteomes of yeast diluted in a HeLa background at five ratios. We find that quantitative measurements made by EncyclopeDIA agree closely with the expected ratios and we now present this result as a new Figure 5.

Q3) How does the performance of the new method compare to the recently described strategy by Bruderer et al. (PMID: 29070702)

The optimized DIA method demonstrated by Bruder et al {Bruderer et al., 2017, #30460} requires an on-column DDA-based spectrum library. For example, the library generated for that paper requires 30 DDA injections of offline high-pH reverse phase (RP) and strong anion exchange SAX fractionated samples. This approach produces an impressive library with instrument-specific retention times, at the expense of time, sample, and significant effort offline fractionating the sample. Bruder et al demonstrate that project-specific DDA libraries significantly outperform the Pan-Human library {Rosenberger et al., 2014, #70337} in the number of peptide detections (1.67x for HEK-293 and 2.05x for triplicate HeLa experiments).

Our goal is to demonstrate that we can reuse previously acquired libraries to avoid recollecting project-specific DDA libraries for every experiment. Our main advance is that using only 6 gas-phase fractionated runs, we can “calibrate” previously acquired DDA libraries to our

instrument platform, regardless of origin. Gas-phase fractionation is much easier and reproducible than offline high-pH RP or SAX fractionation because it's performed directly in the mass spectrometer, and doesn't require additional chromatographic setup. Ultimately, our approach enables the reuse of DDA libraries, even across laboratories and instrument platforms. In the first submission we showed that we could use cross-lab DDA spectrum libraries to build a chromatogram library (Figure 2a). In this revision we now demonstrate analogous results by rebuilding our chromatogram library with the Pan-Human library. Despite the fact that this library was generated in the Aebersold Laboratory using Sciex Q-ToFs, we calibrate it in both retention time and fragmentation patterns to our QE-HF and improve detection rates. We have added this new result in Figure 2a:

Reviewer #3 (Remarks to the Author):

In their manuscript "Comprehensive peptide quantification for data independent acquisition mass spectrometry using chromatogram libraries", Searle et al. present a new workflow for identification and quantification of DIA data based on chromatogram libraries. DIA data has gained increasing traction in the last few years especially through the development of SWATH-MS and the development of new tools and approaches in DIA data analysis is important in the field. The proposed study is an interesting contribution to the field as it combines multiple approaches previously described into a new workflow and describes an open-source tool to run the data analysis. The novelty of the method lies in a novel way of constructing spectral libraries for peptide-centric DIA analysis and incremental improvements to the already published approaches for peptide-centric analysis.

While the development of novel methods for DIA data analysis is of high interest, the current manuscript lacks in several major aspects. First and foremost, it is unclear whether the major advance described here is a new tool or a new way to construct spectral libraries. While the development of a "chromatographic library" seems to be the main advance, the authors remain surprisingly vague about what exactly their "chromatographic library" really contains and how it differs from traditional assay libraries. It is also not clear whether the new "chromatographic libraries" are actually performing better than traditional assay libraries since the authors fail to make an appropriate comparison. Another major question regarding the new library generation method is whether the authors employ appropriate control of the false discovery rate when constructing the library. Importantly, the authors should put their work in proper context with previous work, citing the appropriate papers where they draw their inspiration for their improvements from, they discuss and cite a lot of work from the MacCoss lab but fail to mention other relevant work. Finally, the authors produce a new tool for DIA data analysis called EncyclopeDIA which uses the new assay libraries, but fail to compare it to existing approaches.

Major comments:

- The authors propose a new method for identifying peptides and generating spectral libraries directly from DIA data. They report a fantastic increase of almost 2-fold in the number of peptides in their narrow-window DIA identification compared to their previous tool using Walnut / PECAN. To achieve this, the authors used a spectral library search, however the authors do not adequately describe how they achieved spectral library search in DIA data and how they perform error control in DIA datasets. The authors should (i) describe their approach in detail and (ii) report how many identifications tools built for spectral library searches in DIA data (such as MSPLIT-DIA) report. Since all of their subsequent results rest on their initial spectral library with 99.6k unique peptides, I suggest to better describe the library generation procedure and validate their findings of 99.6k peptides with MSPLIT-DIA or SpectraST.

We thank the reviewer for this suggestion and we have expanded the “chromatogram library generation” section from one paragraph to four. Now we describe the contents of the libraries and the library generation procedure in much greater depth. Briefly, chromatogram libraries provide a calibration step that substitutes the data in DDA spectrum libraries or fragmentation models in database search engines for DIA-specific fragmentation and HPLC/column-specific retention times. In addition, chromatogram libraries contain peptide peak shape and an indication of fragment ions that have expected interferences from the narrow-window data. Due to gas-phase fractionated tiling, each peptide is only represented in the narrow-window data once, which eliminates the need for spectrum averaging (e.g. SpectraST) or best spectrum selection (e.g. Bibliospec) typically used by DDA-based library curation tools. We now include a direct comparison to the library filtering criteria in SpectraST beyond simple peptide-level FDR thresholding. While the MSPLIT-DIA library search algorithm does not include a library generation approach, we now refer to the algorithm in the introduction. Finally, the difference in chromatogram library size by searching a DDA-based spectrum library (166k unique peptides) with EncyclopeDIA or a FASTA database (1143k unique +2H/+3H tryptic peptides) with Walnut is in part because the spectrum library represents a more targeted search space, while additionally including expected post-translationally modified (oxidized and acetylated) peptides, as well as peptides with multiple missed cleavages and expected +4H/+5H/+6H peptides.

- The authors should cite appropriate literature describing similar advances in the field and acknowledge previous achievements in the field. For example, they are not the first ones to report better identification and quantification in DIA compared to DDA (this has been consistently reported since the first high throughput peptide-centric DIA papers were published). Also their approach of RT normalization seems similar to the high-precision iRT approach by Bruderer et al (Proteomics, 2016). The authors should also cite and contrast their work on fragment ion interference removal to recent work such as mapDIA and SWATHProphet that essentially aim to achieve something similar.

We thank the reviewer for these references and now cite the high-precision iRT approach, mapDIA, and SWATHProphet in the manuscript. We have also substantially increased the number of citations throughout the manuscript, particularly when discussing improved identification and quantification rates in DIA compared to DDA. We agree that this is not the novelty of our work: here we demonstrate that these improvements are possible from just collecting DIA data alone without the need to generate large scale DDA spectrum libraries.

The retention time mapping algorithm in EncyclopeDIA is similar to the Bruderer et al high-precision iRT method {Bruderer et al., 2016, #21583} and we appreciate this reference. We include the following text when discussing retention times: “Retention times in aggregate spectrum libraries are typically derived by linearly interpolating multiple DDA data sets to a known calibration space (such as that defined by the iRT standard {Escher et al., 2012, #80751}), which enables retention times to be comparable from run to run, or even across platforms. However, these measurements usually contain some wobble due to errors introduced by assuming a linear fit. Bruderer et al {Bruderer et al., 2016, #21583} improve upon this

strategy using high-precision iRT fitting using a non-parametric curve fitting approach for hundreds or thousands of peptides, and EncyclopeDIA uses an analogous kernel density estimation approach to fit retention times between wide-window DIA results and retention times in libraries.”

The Bruderer et al method (high-precision iRT fitting, or HP-iRT) works by producing hundreds or thousands of bins and calculates local Theil-Sen linear regressions to generate x-y reference points. In comparison, the alignment results produced by EncyclopeDIA differ with those from HP-iRT in the following ways:

- 1) During DDA library construction tens or hundreds of raw files are aligned. Errors in DDA library construction can affect the median retention time points used by Theil-Sen regressions, which can sometimes produce erroneous fits. In contrast, the EncyclopeDIA algorithm only tracks the local maxima and is not influenced by any other groups of retention time points
- 2) Unlike the HP-iRT method, the EncyclopeDIA algorithm does not permit micro adjustments that allow for “negative” retention time. The EncyclopeDIA algorithm guarantees a monotonic fit that always increases with time
- 3) The EncyclopeDIA algorithm uses density of points to draw curves, and consequently performs poorly when the number of detected peptides drops below low N. Consequently, below N=20 peptides EncyclopeDIA resorts to linear regression. In this case, the HP-iRT method shrinks the number of bins, which may produce a more natural fit as N decreases

To demonstrate the benefits of both points 1 and 2, we investigated curve fitting against a previously published iRT library generated by Skyline{Lawrence et al., 2016, #57866}. In this DDA-based library it appears at least one input file was improperly aligned during library construction. This flaw in library construction is noticeable when the DDA library is aligned to a wide-window DIA experiment, resulting in streaking in the iRT dimension. In this case, while the HP-iRT method is thrown off by this streaking, the KDE method in EncyclopeDIA still produces an appropriate curve fit:

We have not added this figure into the manuscript but are happy to do so if the reviewer deems it useful.

- It is currently unclear what the "chromatographic library" generated here really entails and what its novel aspect is. Currently the main advance seems to be the fact that the retention times in the library are more accurate than in previous, iRT based libraries since they are derived on the same column. Also the authors speculate that the fragment ion intensities may be more accurate. However, the authors need to clarify how their approach differs from the high-precision iRT concept (Bruderer et al, Proteomics, 2016) and whether their "chromatographic library" features any additional improvements compared to the "high precision iRT library" described in this publication.

As stated above, we have expanded the "chromatogram library generation" section to add more clarifying details. Our goal is to demonstrate that we can reuse previously acquired global DDA libraries to avoid recollecting project-specific DDA libraries for every experiment. Rather than requiring offline fractionated, column-specific DDA libraries for each new experiment, we propose using 6 gas-phase fractionated DIA runs that "calibrate" previously acquired DDA libraries to our instrument platform. As stated above, in this revision we demonstrate that by generating a chromatogram library, we can reuse the Pan-Human library {Rosenberger et al., 2014, #70337} to achieve detection rates within 93% of our sample-specific HeLa library. This advance enables recycling previous previous efforts in the literature to generate deep DDA libraries and significantly lowers the barrier of entry for scientists to analyze DIA data.

While accurate retention times are a significant benefit of using a DIA-based calibration step, fragmentation pattern accuracy also improves. We demonstrate that the DIA-based chromatogram library fragmentation patterns are more representative of wide-window DIA patterns than DDA-based spectrum libraries in Figure 3e and 3f. Here we calculate the distribution of correlation coefficients between the DDA library (3e) and DIA library (3f) with wide-window fragment patterns for B- and Y-type ions, and show that the DIA library produces higher correlation coefficients. In this revision we've scaled the frequency axes to make this comparison more apparent:

We acknowledge that this issue is likely largely resolved if DDA libraries are generated with DIA in mind (i.e. using fixed collision energies that do not adjust based on charge state). However, most current DDA libraries (such as the Pan-Human library) have not been generated in this manner.

- The authors fail to compare their new tool with existing tools. The authors have access to the Skyline tool developed in the same lab which has recently shown to perform equivalent to other tools (OpenSWATH, Spectronaut, SCIEX etc. see Navarro et al 2016). Without proper benchmarking and tool comparison, the readers are left in the dark whether the new tool / workflow provides a substantial improvement over existing approaches. It seems straight forward for example to load the 99.6k (or a subset thereof) peptide library into either Skyline, OpenSWATH or Spectronaut and perform peptide-centric analysis. This direct comparison would quantify the advantage that the new "chromatographic libraries" approach brings in conjunction with existing tools like Skyline.

We have added a new Skyline analysis and present that as Supplementary Figure 1. Rather than running the risk of misrepresenting other developers tools, we have asked Brendan MacLean to join our author list and perform this analysis. We find that the number of peptide detections at 1% peptide FDR in triplicate HeLa injections consistently improve by 1.15x with Skyline when using the HeLa chromatogram library compared to searching the HeLa DDA-based spectrum library.

- I seem to not find a comparison of the new library generation approach to the traditional library generation approach. Maybe I missed it and the authors can point me to the appropriate figure, but it seems straight forward to use a large scale library such as the 166.4k peptide library the authors have access to and directly search the wide window DIA data. This would showcase whether the somewhat labour-extensive step of intermediately creating narrow window DIA data to create a 2nd library is necessary and improves over the straight forward approach of directly using the initial 166.4k peptide library for peptide-centric extraction. If this is what Figure 2a shows, please be more clear about this.

The reviewer is correct that Figure 2a/d shows a comparison between the new chromatogram library approach and the traditional spectrum library approach. We've tried to clarify this figure with additional structure. Colored shading indicates binary comparisons:

- light orange indicates searching a FASTA versus a FASTA-based chromatogram library
- light purple indicates searching a traditional sample-specific spectrum library versus a chromatogram library built from that spectrum library
- and light green indicates searching a global organism-specific spectrum library versus a chromatogram library built from that spectrum library

In all cases we see an improvement in the number of overall peptide detections. In addition to clarifying the comparisons, we have made the library names more consistent. The updated figure is:

- The authors claim that their fragment ion interference removal provides improved quantitative accuracy. However, they do not provide evidence for such a claim. Either they need to remove the claim or show data to back this up - e.g. show improved CV / quantitative accuracy when turning on their interference removal.

Rather than further complicate this manuscript, we have removed the claim here and will report on it in a future work.

- In addition to citing PECAN as a library-free tool the authors should also cite Spectronaut and DIA-Umpire.

We now cite DIA-Umpire and mention Spectronaut Pulsar by name in the introduction. While to our knowledge Biognosys has not produced a direct citation describing Spectronaut in detail, we had already cited the first use of it in the literature by members of their team (Bruderer et al, 2015{Bruderer et al., 2015, #63308}).

- It is not clear how the authors calculate "global peptide FDR across all experiments" - did they use the method described in ref 18?

In the discussion of Rosenberger et al 2017{Rosenberger et al., 2017, #70407} the authors suggest "that a practical method to control the error rate is to filter the result matrices generated from either the run-specific or experiment-wide contexts using the set of analytes that are confidently detected in the global context" and go on to say that they have shown that "this results in a uniform set of inferred proteins with negligible accumulation of false positives over a large number of samples".

In our manuscript we have attempted to follow this suggestion as closely as possible. When considering a run-specific context, we perform FDR filtering on each run individually at the 1% peptide-level for peptide benchmarking (e.g. Figure 2), or at both the 1% peptide-level and 1% protein-level (after protein grouping parsimony) when considering protein detections (e.g. Figure 3). Since these analyses are performed independently, we do not use any run-to-run alignment. However, when considering an experiment-wide context (e.g. the biological study in Figure 6) we only report analytes detected after both global 1% peptide-level and 1% protein-level FDR filtering using feature sets from all runs simultaneously. We strongly believe in the approach presented in Rosenberger et al 2017 and we have engineered the EncyclopeDIA software such that if run-to-run alignment is performed, global FDR assessment is automatically triggered. We have tried to clarify this point in the main text by more clearly stating what type of FDR correction is used in each analysis.

- The authors should describe more of their experimental setup in the main paper, e.g. gradient length and instrument used when describing the number of identified peptides -- or number of samples and biological / technical replicates used when describing their biological study.

We collected all the DIA runs with a Thermo Q-Exactive HF tandem mass spectrometer using a 90 minute linear gradient. For the biological study we selected six starvation times to match commonly used protocols and generated six biological replicates per condition. Four biological replicates were used to monitor the HeLa phosphoproteomes. We have added these details to the main manuscript.

Minor comments:

- typo in decoy library method: "insuring" should be "ensuring"

This has been corrected.

Reviewers' Comments:

Reviewer #2:

Remarks to the Author:

The authors have – in my opinion - correctly and satisfactorily addressed the concerns & remarks of all three reviewers.

The manuscript has improved considerably in the present revision and is now suitable for publication.

Reviewer #3:

Remarks to the Author:

In their revised manuscript, Searle et al have addressed and clarified many of the reviewer's comments and invested substantial work to perform additional experiments and analysis. Specifically, they have clarified their workflow and have made efforts to put their work in context with other previous research.

Still some work remains to be done, the authors should now discuss their tool in the light of the new findings and with respect to results achieved by Skyline and by Bruderer et al -- in order to allow the reader to understand where the reported gains come from and how they were achieved.

- The authors need to tone down their claims of vast improvements and specifically re-visit their claim of 50% more identified peptides. This improvement is mainly seen with their tool whereas Skyline when presented with the same two input libraries produces a merely 15% increase in peptide identifications. It seems that claim comes from the fact that EncyclopeDIA is particularly bad at dealing with uncalibrated libraries, only identifying 48k peptides whereas Skyline finds 58k peptides even with the uncalibrated library. Thus, the increase is more impressive with EncyclopeDIA (48k to 72k) while Skyline has a modest gain (58k to 67k). Still, it is interesting that EncyclopeDIA produces slightly better results with the calibrated library than Skyline but the gain relative to Skyline of 7% is not extremely impressive. Viewed in this light, the authors approach is indeed able to improve peptide identifications by 15% through better libraries and they add another 7% due to optimizing their tool. However, this improvement is far from the claimed 50% improvement, which seems mainly to be due to the bad performance of EncyclopeDIA on the non-calibrated libraries. In conclusion, the authors should remove their claim of 50% improved identifications and when discussing Figure 2 put these results in context and explain how (and why?) their tool performs badly on uncalibrated libraries.
- Therefore, the authors should clearly state that the calibration of the library enables a gain of about 15% while an additional gain of 7% can be achieved when the full information present in the chromatogram library is used.
- Thus I think the authors should remove / tone down statements such as "One of the primary reasons on-column chromatogram libraries enable such high performance ... [is improved RT]" especially as these claims are (i) not *directly* supported by data (the authors would have to show how much of the gain is due to improved RT) and (ii) the actual gain is rather modest.
- The authors speculate where the gains in their library potentially come from by showing improved RT and fragmentation library pattern correspondence. It would be more convincing if the authors actually showed data to support this claim (e.g. by doing the calibration only on the fragmentation patterns and not the RT or the other way round).
- The authors should discuss their gains in peptide id in the context of the peptide id gains reported by Bruderer et al (do they gain more than Bruderer from accurate RT or a similar amount?)
- Similarly, the claims of vast improvements compared to Walnut should be discussed in the context of Walnut finding much fewer peptides than even a regular DDA approach.
- The authors should comment whether their selection of 3 interference free transitions is ad-hoc or based on simulations / theory to be an appropriate number of transitions for unambiguous

peptide identifications (it seems rather low).

- Data deposition: the authors need to deposit their data on an official site that is associated with proteomexchange (such as PRIDE) instead of their own in-house Chorus server.

Minor points:

- A suggestion: I believe that the manuscript loses some of its focus / story line due to the fact that the authors provide two competing workflows: one based on PECAN / Walnut and library free and a second one based on re-calibrating spectral libraries. The authors may consider to lay the main focus on one of the two workflows and present a more streamlined story to the reader. Due to this fact, the text sometimes jumps from one workflow to the other, leading to some confusion on the side of the reader.

- I think it would help a lot if in Figure 1 it became more clear that the "on column chromatogram library" is a tuned and calibrated spectral library that is then used for targeted DIA extraction

- I suggest to change the title, since there is no evidence that the authors perform "comprehensive peptide quantification" (where do the authors show comprehensiveness?) and rather change it to something more adequate describing how chromatogram libraries calibrate spectral libraries for improved accuracy.

First, we would like to thank the reviewer for positively receiving our updated manuscript and for providing additional comments. We have attempted to thoroughly address all of these comments.

In their revised manuscript, Searle et al have addressed and clarified many of the reviewer's comments and invested substantial work to perform additional experiments and analysis. Specifically, they have clarified their workflow and have made efforts to put their work in context with other previous research.

Still some work remains to be done, the authors should now discuss their tool in the light of the new findings and with respect to results achieved by Skyline and by Bruderer et al -- in order to allow the reader to understand where the reported gains come from and how they were achieved.

- The authors need to tone down their claims of vast improvements and specifically revisit their claim of 50% more identified peptides. This improvement is mainly seen with their tool whereas Skyline when presented with the same two input libraries produces a merely 15% increase in peptide identifications. It seems that claim comes from the fact that EncyclopeDIA is particularly bad at dealing with uncalibrated libraries, only identifying 48k peptides whereas Skyline finds 58k peptides even with the uncalibrated library. Thus, the increase is more impressive with EncyclopeDIA (48k to 72k) while Skyline has a modest gain (58k to 67k). Still, it is interesting that EncyclopeDIA produces slightly better results with the calibrated library than Skyline but the gain relative to Skyline of 7% is not extremely impressive. Viewed in this light, the authors approach is indeed able to improve peptide identifications by 15% through better libraries and they add another 7% due to optimizing their tool. However, this improvement is far from the claimed 50% improvement, which seems mainly to be due to the bad performance of EncyclopeDIA on the non-calibrated libraries. In conclusion, the authors should remove their claim of 50% improved identifications and when discussing Figure 2 put these results in context and explain how (and why?) their tool performs badly on uncalibrated libraries.

The reviewer brings up several important points and we sought to understand this behavior better with some new analyses. While it is true that EncyclopeDIA performs worse than Skyline using the HeLa-specific spectrum library, we now find that they both EncyclopeDIA and Skyline perform equally well with the Pan-Human spectrum library. We suspect this variation is due to EncyclopeDIA's reliance on accurate retention time stability. To corroborate this, we found that our HeLa-specific library has higher retention time spread than the Pan-Human library, mainly due to the fact that the HeLa-specific library runs were not originally collected for this purpose. Since most spectrum

libraries are built-for-purpose, we feel that the Pan-Human library performance is more indicative and that in most cases, EncyclopeDIA and Skyline should perform equally well with DDA-based spectrum libraries.

We would like to point out that achieving our high level of success with Skyline is not trivial. Significant iterative tuning of the normalized retention time library, seed peptides, and search parameters was required. For example, our first-pass analyses before adding Brendan MacLean as a collaborating author produced approximately 2/3rds the number of peptides, despite the fact that our group has significant experience processing DIA experiments with Skyline. Meanwhile, EncyclopeDIA produces this level of performance “out of the box”.

Regardless, in both cases, Skyline shows approximately a 15% increase using chromatogram libraries over the original DDA-based spectrum libraries. Using chromatogram libraries, EncyclopeDIA produces a consistent 20-25% peptide detection improvement over Skyline using DDA libraries. We have adjusted the abstract and discussion of Figure 2 to present this result. Additionally, we have updated Supplementary Figure 2 to include this new analysis:

- Therefore, the authors should clearly state that the calibration of the library enables a gain of about 15% while an additional gain of 7% can be achieved when the full information present in the chromatogram library is used.

Please see the point above, and our adjusted abstract and discussion.

*- Thus I think the authors should remove / tone down statements such as "One of the primary reasons on-column chromatogram libraries enable such high performance ... [is improved RT]" especially as these claims are (i) not *directly* supported by data (the*

authors would have to show how much of the gain is due to improved RT) and (ii) the actual gain is rather modest.

As the reviewer suggested, we have toned down this language. However, this comment inspired us to investigate the impact of retention time and fragmentation patterns independently. As suggested in the comment below, we have searched the HeLa benchmarking datasets using the HeLa-specific chromatogram library (DIA), where either the retention times or fragmentation patterns have been switched with the spectrum library (DDA). Compared to the original HeLa-specific spectrum library search (average of 47863 peptides), a small improvement (3%) comes from simply using a narrowed peptide selection. DIA-based retention times and fragmentation patterns provide a 22% and 21% improvement over this, respectively. Using both DIA-based retention times and fragmentation patterns provides a 46% improvement, suggesting that these two factors are additive. We have added the following new Supplementary Figure 3 describing this phenomenon:

- The authors speculate where the gains in their library potentially come from by showing improved RT and fragmentation library pattern correspondence. It would be more convincing if the authors actually showed data to support this claim (e.g. by doing the calibration only on the fragmentation patterns and not the RT or the other way round).

Please see the point above, and our adjusted abstract and discussion.

- The authors should discuss their gains in peptide id in the context of the peptide id gains reported by Bruderer et al (do they gain more than Bruderer from accurate RT or a similar amount?)

Bruderer et al 2016{Bruderer et al., 2016, #21583} demonstrate that high-precision iRTs provide up to 23% improvement in the number of detected peptides. Although not measuring the same source of accurate retention times, our findings (22% improvement due to column-specific retention times) agree closely with this result. Unfortunately a direct comparison with Bruderer et al 2017{Bruderer et al., 2017, #30460} regarding the number of detected peptides is impossible due to significant sample loading and chromatographic differences. Specifically, in our experiments we loaded 1 ug sample with 30 cm columns and 90 minute linear gradients, while Bruderer et al 2017 loaded up to 4 ug sample with 1 m columns and 240 minute linear gradients. However, we can discuss the results presented by Bruderer et al 2017 relative to our work in general terms.

Bruderer et al 2017 demonstrate significantly improved detection rates when searching project-specific spectral libraries as opposed to global libraries like the pan-human library. As discussed above, this is likely due to three factors:

1. Lowering the number of peptide queries softens false discovery correction by asking fewer questions
2. Retention times are tuned to a specific HPLC setup
3. Fragmentation patterns are tuned for a specific mass spectrometer

Our analysis enables us to differentiate between these three factors within the context of our experiments: we find that improving fragmentation accuracy for DIA acquisition is approximately as important as improving retention time accuracy and that both improvements in tandem make up the large majority of our reported gains.

One major complication of the Bruderer et al 2017 approach is that a new spectral library needs to be created for each new chromatographic condition. Specifically, the entire library must be recreated if a column or trap needs to be replaced during an experiment. Creating these libraries is difficult (e.g. Bruderer et al 2017 use 30 DDA injections of both offline high-pH reverse phase and strong anion exchange fractionated samples) and as proteomics experiments get larger and larger, column and trap clogs or instability become more common. The main benefit of our approach is that rather than recollecting an entire spectral library for each new chromatographic condition, we only collect six new injections to calibrate an old library. We have added additional emphasis to help clarify this point in the abstract.

- Similarly, the claims of vast improvements compared to Walnut should be discussed in the context of Walnut finding much fewer peptides than even a regular DDA approach.

We have added the following underlined context to this discussion: “We also evaluated the creation of chromatogram libraries using a DIA-only workflow. Using this approach, we were able to detect an average of 20.6k peptides from the Uniprot Human FASTA database using Walnut, or approximately 0.6x of the detections found by top-20 DDA. In contrast, we found an average of 47.8k peptides (2.3x increase) when we searched the Walnut-based chromatogram library with EncyclopeDIA (Figure 2a), or approximately 1.4x more than DDA. These results agree with previous work{Ting et al., 2017, #29966} showing that Pecan does not perform as well as DDA when searching wide-window runs, but typically outperforms DDA when searching gas-phase fractionated runs.”

- The authors should comment whether their selection of 3 interference free transitions is ad-hoc or based on simulations / theory to be an appropriate number of transitions for unambiguous peptide identifications (it seems rather low).

While using a minimum of three transitions for quantification is common in the SRM community (e.g. Prakash et al, J Proteome Res. 2009 Jun;8(6):2733-9), we have explored the effect of number of transitions on peptide quantification reproducibility. Using the triplicate HeLa injections, we compared the coefficient of variation for peptides quantified with different numbers of transitions. The median CV for peptides with three transitions is 24.6%, which we feel is acceptable for this study. We break this down in the following box and whisker plot, where boxes indicate medians and interquartile ranges, and whiskers indicate 5% and 95% values:

This measurement will change depending on the background, and in certain circumstances it might make sense to raise this or lower the number of required transitions. Indeed, EncyclopeDIA supports different options for number of required transitions. We have not added this experiment (or figure) into the manuscript but are happy to do so if the reviewer deems it useful.

- Data deposition: the authors need to deposit their data on an official site that is associated with proteomexchange (such as PRIDE) instead of their own in-house Chorus server.

We have uploaded all raw data files discussed in this manuscript to MassIVE and have already made it public. We have added the following text to the software/data availability section: "All mass spectrometry mzML and RAW data files (see Supplementary Table 3 for raw data annotations) are available on the Chorus Project (project identifier 1433) and at the MassIVE proteomics repository (project identifier MSV000082805)."

Minor points:

- A suggestion: I believe that the manuscript loses some of its focus / story line due to the fact that the authors provide two competing workflows: one based on PECAN / Walnut and library free and a second one based on re-calibrating spectral libraries. The authors may consider to lay the main focus on one of the two workflows and present a more streamlined story to the reader. Due to this fact, the text sometimes jumps from one workflow to the other, leading to some confusion on the side of the reader.

We agree and we have changed this order in the discussion. We now present a DDA/DIA workflow first, and a DIA-only workflow second.

- I think it would help a lot if in Figure 1 it became more clear that the "on column chromatogram library" is a tuned and calibrated spectral library that is then used for targeted DIA extraction

We respectfully disagree with the reviewer on this point, since Figure 1 also shows a DIA-only workflow with Walnut, which does not use spectrum libraries at all. Therefore, referring to a chromatogram library as a "calibrated spectrum library" does not tell the complete narrative.

- I suggest to change the title, since there is no evidence that the authors perform "comprehensive peptide quantification" (where do the authors show

comprehensiveness?) and rather change it to something more adequate describing how chromatogram libraries calibrate spectral libraries for improved accuracy.

We have changed the title to “Chromatogram libraries improve peptide detection and quantification by data independent acquisition mass spectrometry”.

Reviewers' Comments:

Reviewer #3:

Remarks to the Author:

The authors have substantially revised their manuscript to my satisfaction. The manuscript can be published as-is

Minor points:

- It may be illustrative to include a description of the "significant tuning" of Skyline necessary to achieve the results displayed in the paper. I could not find any description regarding this process in the paper.
- It is also nice to see that the improvement reported here corresponds well with the percentage improvement reported by Bruderer et al

Reviewer #3 (Remarks to the Author):

The authors have substantially revised their manuscript to my satisfaction. The manuscript can be published as-is

Minor points:

- It may be illustrative to include a description of the "significant tuning" of Skyline necessary to achieve the results displayed in the paper. I could not find any description regarding this process in the paper.

We agree and have included a new Supplementary Note 2 that outlines how Skyline was run to achieve the results in the manuscript. In general we followed the "Large Scale DIA with Skyline" webinar (<https://skyline.ms/webinar14.url>), which we link to in the manuscript.

- It is also nice to see that the improvement reported here corresponds well with the percentage improvement reported by Bruderer et al

We agree that this was a nice confirmatory observation and we thank the reviewer for encouraging us to consider this line of reasoning.